


# MOIST: a MATLAB-based fully coupled one-dimensional isotope and soil water transport model

Han Fu[1], Eric J. Neil[1], Huijie Li[2], Bingcheng Si[1,2]

[1]Soil Science Department, University of Saskatchewan, Saskatoon, S7N 5A8, Canada

[2]College of Resources and Environmental Engineering, Ludong University, Yantai, Shandong Province, 264025, China

*Correspondence to:* Bingcheng Si (bing.si@usask.ca)

**Abstract.** Modeling water stable isotope transport in soil is crucial to sharpen our understanding of water cycles in terrestrial ecosystems. However, isotope and soil water transport are not fully coupled in current models. In this study, we developed MOIST, a MATLAB-based one-dimensional isotope and soil water transport model, a program that solves one-dimensional

water, heat, and isotope transport equations simultaneously. Results showed that the MOIST model has good agreements to the theoretical tests and semi-analytical solutions of isotope transport under fixed boundary conditions. Furthermore, we validated the program with short- and long-term measurements from lysimeters studies. The overall Nash-Sutcliff efficiency coefficient (*NSE*) of soil water and deuterium ([2]H) transport for the short-term measurements are 0.66 and 0.69, respectively, with respective determine coefficient ($R^2$) of 0.82 and 0.70, mean absolute error (*MAE*) of 0.02 $m^3$ $m^{-3}$ and 11.84‰. For the

long-term lysimeter study, the overall *NSE*, $R^2$, and *MAE* of simulated $\delta^{18}O$ are 0.47, 0.49, and 0.92‰, respectively. These indices indicated the excellent performance of the MOIST model in simulating water flow and isotope transport in simplified ecosystems, suggesting a great potential of our program in promoting understandings of ecohydrological processes in terrestrial ecosystems.

**1 Introduction**

Hydrogen and oxygen stable isotopes are powerful tools to reveal and document water cycles and ecohydrological processes in terrestrial ecosystems (Vereecken et al., 2016). Their applications encompass plant water sourcing (Brooks et al., 2010), evaporation estimation (Walker et al., 1988; Xiang et al., 2021), changes in soil water storage, water age identification (Hoffmann et al., 2004; Jouzel et al., 2007; Zhang et al., 2019), and evapotranspiration partitioning from plots to continental

scales (Evaristo et al., 2015; Vereecken et al., 2016). However, despite significant advances in experimental techniques and in-situ sampling of water isotopes in terrestrial ecosystems (Jasechko et al., 2013), a key problem remains unsolved: existing models cannot accurately characterize the fractionation process, i.e., the transfer of water isotopes in vapor between the atmosphere and soil. (Zhou et al., 2021). This limits our ability to accurately simulate isotope transport in soil-vegetation-atmosphere systems and to sharpen our understanding of evaporations and transpiration on an ecosystem scale (Vereecken et

al., 2016; Beyer et al., 2020).

A few attempts have been made to develop numerical models of water stable isotope transport in soil. Based on Melayah et al., 1996a, Braud et al. (2005) developed the "SiSPAT-Isotope" (Simple Soil-Plant-Atmosphere Transfer) model, which incorporates the resistance to isotope transport between soil surface and atmosphere (Braud, et al., 2009a; Braud, et al., 2009b).

Subsequently, Haverd and Cuntz (2010) developed "Soil-Litter-Iso", which is also a one-dimensional model for the transport of heat, water, and stable isotopes in soil containing surface litter and actively transpiring vegetative cover. This model extended the linearization method from Ross (2003) to vapor transport. Compared to SiSPAT-Isotope, Soil-Litter-Iso is more efficient for thicker soil layers and larger time steps because of the implementation of linearization in the model. However, Soil-Litter-Isotope does not consider liquid and vapor heat capacity variation and the changes in vapor volume for heat transport, which

may substantially bias the heat flux transport within the soil (Satio et al., 2006), resulting in biased isotope transport fluxes.

Another modeling approach capitalizes on the capability of the well-known model, HYDRUS-1D, for modeling water flow and chemical transport. Stumpp et al. (2012) simulated $^{18}$O movement in soil by modifying a solute transport module of HYDRUS-1D. However, Stumpp et al. (2012) neglected fractionation and so their model is only applicable in situations where

the slope of the isotopic evaporation line from soil water is close to the local meteoric water line (Stumpp et al., 2012). The latest version of HYDRUS-1D has fixed this problem by integrating the SiSPAT-Isotope model (Zhou et al., 2021). However, all these modelling approaches consider water and isotope transport separately, where isotope transport equations are solved only after solving water and heat equations at each time step. As a result, they require the mass balance errors of soil water to be extremely small in the magnitude of $10^{-16}$ m s$^{-1}$ (Zhou et al., 2021), which demands costly high spatial resolutions (Braud

et al., 2005) or advanced discretization schemes (Haverd and Cuntz, 2010; Zhou et al., 2021) that require longer computational times. Thus, efficient models that simultaneously solve soil water, heat, vapor, and isotope transport are urgently needed.

Therefore, the objectives of this study are: (1) to develop a one-dimensional model which can solve fully coupled soil water, heat, vapor, and isotope transport simultaneously and (2) to validate the model through theoretical tests, analytical solutions

for specific boundary conditions and field measurements. The model is expected to be efficient for simulating isotope transport within soil under deeper and longer spatial and temporal scales than existing models by solving fully coupled water flow, heat, vapor, and isotope transport equations. To increase accessibility, the program is written in MATLAB language. We describe the model formulation and boundary conditions below, which are followed by the illustration of efficiency and accuracy under theoretical, semi-analytical, and field validations.



## 2 Material and Methods

### 2.1 Model description

The MOIST model solves soil water, heat, and isotope transport equations simultaneously (Fig. 1). The processes are described in detail in the order shown in the diagram (Fig. 1).

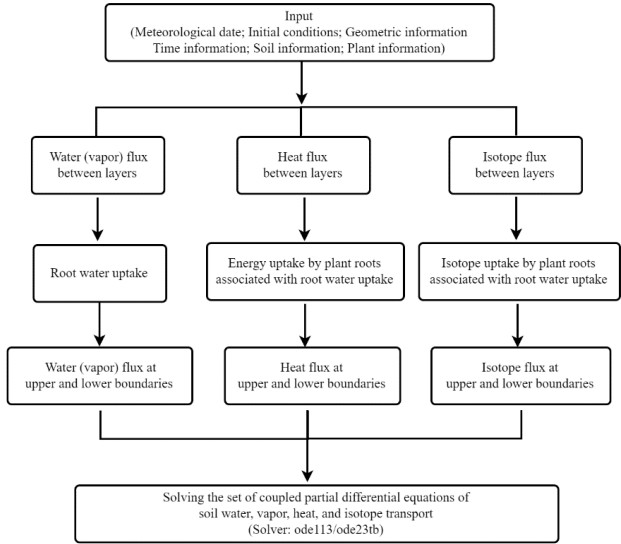

**Figure 1. The framework diagram of the MOIST model.**

### 2.1.1 Soil water and heat transport

One-dimensional water and heat transport within soil can be described by mass and energy conservation equations with a downward-positive coordinate system (Banimahd and Zand-Parsa, 2013; Haverd and Cuntz, 2010):

$$\frac{\partial(\theta + \theta_v)}{\partial t} = -\frac{\partial(q_l + q_v)}{\partial z} - S_p \tag{1}$$

$$C_{soil}\frac{\partial T}{\partial t} = -\frac{\partial\left(-K_H\frac{\partial T}{\partial z} + \rho\lambda_E q_v\right)}{\partial z} \tag{2}$$

where $\theta$ (m³ m⁻³) and $\theta_v$ (m³ m⁻³) are the volumetric soil water and vapor content; $t$ (s) is time; $q_l$ (m s⁻¹) is the liquid flux and $q_v$ (m s⁻¹) is the vapor flux; $z$ (m) is the spatial distance; $S_p$ (s⁻¹) is the sink term, which is zero when there is no sink during simulation (e.g. root water uptake); $C_{soil}$ (J m⁻³ K⁻¹) is the soil volumetric heat capacity; $K_H$ (W m⁻¹ K⁻¹) is the soil thermal conductivity; $T$ (K) is the temperature; $\rho$ (kg m⁻³) is the water density and; $\lambda_E$ (J kg⁻¹) is the latent heat of vaporization. The liquid flux, $q_l$ (m s⁻¹), can be calculated by Darcy's law (positive downwards):

$$q_l = -K\frac{dh}{dz} + K \tag{3}$$

where $K$ (m s⁻¹) is the unsaturated hydraulic conductivity and $h$ (m) is the pressure head. The vapor flux, $q_v$, can be calculated



by Fick's law:

$$q_v = -D_v \frac{dc_v}{dz} \tag{4}$$

where $Cv$ (m³ m⁻³) is the vapor concentration in soil air space, which is the product of saturated vapor concentration ($Cv_{sat}$)

and relative humidity in soil ($h_r$) (Haverd and Cuntz, 2010). Considering the influence of liquid and vapor heat capacity

variation on heat transport, Eq. (2) is extended to (Saito et al., 2006; Šimůnek et al., 2013):

$$C_{soil} \frac{\partial T}{\partial t} + \lambda_E \frac{\partial \theta_v}{\partial t} = -\frac{\partial \left( -K_H \frac{\partial T}{\partial z} + \rho \lambda_E q_v + C_w T q_l + C_{vh} T q_v \right)}{\partial z} - C_w S_p T \tag{5}$$

where $C_w$ (J m⁻³ K⁻¹) and $C_{vh}$ (J m⁻³ K⁻¹) are the heat capacities of liquid water and vapor.

**2.1.2 Soil isotope transport**

The abundance of water stable isotopes is conventionally expressed as $\delta$ values, in units of per mil (‰). However, for

convenience, the abundances are presented in concentration, $C_i$ (kg m⁻³). Relationships between isotopic ratio ($R_i$) and

concentration ($C_i$) can be expressed as (Melayah et al., 1996a; Braud et al., 2005):

$$C_i = \frac{M_i}{M_w} R_i \rho \tag{6}$$

where $M_i$ (kg mol⁻¹) is the molar mass for a given isotope species, $i$; $M_w$ (kg mol⁻¹) is the molar mass of water and $\rho$ (kg m⁻³)

is the water density.

The isotope mass conservation equation for both liquid and vapor phases is:

$$\frac{\partial \left( C_i^l \theta + C_i^v \theta_v \right)}{\partial t} = -\frac{\partial (q_i)}{\partial z} - C_i^l S_p \tag{7}$$

where $C_i^l$ (kg m⁻³) and $C_i^v$ (kg m⁻³) are the concentration of isotope species $i$ in liquid and vapor phases; $\theta$ (m³ m⁻³) and $\theta_v$

(m³ m⁻³) are the soil water content and vapor content, respectively. Note that $\theta_v$ in Eq. (1) and (7) is given in terms of an

equivalent water content:

$$\theta_v = Cv_{sat} h_r \theta \tag{8}$$

$q_i$ is the total isotopic flux (positive downwards), which consists of liquid isotopic flux, $q_i^l$, and vapor isotopic flux, $q_i^v$:

$$q_i^l = C_i^l q_l - D_{i,s}^l \frac{\partial C_i^l}{\partial z} \tag{9}$$

$$q_i^v = -D_{i,s}^v \frac{\partial C_i^v}{\partial z} \tag{10}$$

where $D_{i,s}^l$ (m² s⁻¹) and $D_{i,s}^v$ (m² s⁻¹) are the liquid and vapor diffusivity in soil for isotope species $i$, and can be defined as

(Melayah et al., 1996a):

$$D_{i,s}^v = D_{va} \tau (\theta_{sat} - \theta) \left( \frac{D_i^v}{D_{va}} \right)^{n_D} \tag{11}$$

$$n_D = 0.67 + 0.33 e^{\left( 1 - \frac{\theta}{\theta_r} \right)} \tag{12}$$



where $\tau$ is the soil tortuosity; $\theta_{sat}$ (m³ m⁻³) and $\theta$ (m³ m⁻³) are saturated and unsaturated soil water content, respectively. $D_v^i$ (m² s⁻¹) is isotopic vapor diffusivity in air:

$$D_v^i = D_{va}\alpha_{diff} \tag{13}$$

$D_{va}$ (m² s⁻¹) is vapor diffusivity in air and can be calculated by (Philip and de Vries, 1957):

$$D_{va} = \frac{D_{v0}10^{-5}}{P_{atm}}\left(\frac{T}{273.16}\right)^{1.88} \tag{14}$$

where $D_{v0}$ is water vapor diffusivity at 0 K ($2.12\times10^{-5}$ m² s⁻¹); $P_{atm}$ is atmospheric pressure (101hpa), and $T$ (K) is the temperature. $\alpha_{diff}$ is 0.9755 for deuterium and 0.9723 for oxygen-18 (Merlivat, 1978; Haverd and Cuntz, 2010).

Liquid diffusion of isotope species in soil, $D_{i,s}^l$ (m² s⁻¹), is written as:

$$D_{i,s}^l = D_i^l\tau\theta + \Lambda|q_l| \tag{15}$$

where $\Lambda$ (m) is dispersivity length; $D_i^l$ (m² s⁻¹) is the diffusivity of liquid isotope species, which is a factor of temperature (Braud et al., 2005):

$$D_i^l = a'\times10^{-9}e^{\left(-\frac{535400}{(T+273.15)^2}+\frac{1393.3}{T+273.15}+2.1876\right)} \tag{16}$$

where $a'$ is a coefficient equal to 0.9833 for deuterium, and 0.9669 for oxygen-18.


Assuming an instantaneous equilibrium between liquid and vapor phases (Vanderborght et al., 2017), a relationship between liquid and vapor isotopic concentration can be expressed as:

$$C_i^v = \alpha_i^*C_i^l \tag{17}$$

where $\alpha_i^*$ is the equilibrium fractionation coefficient, which can be written as (Braud et al., 2005):

$$\alpha_i^* = e^{\left(\frac{a^*}{(T+273.15)^2}-\frac{b^*}{T+273.15}-c^*\right)} \tag{18}$$

where coefficients $a^*$, $b^*$, $c^*$ are 24844, -76.248, 0.052612 for deuterium, and 1137, -0.4156, -0.0020667 for oxygen-18.

Finally, isotope transport equations can be written as:

$$\frac{\partial\left\{C_i^l[\theta+\alpha_i^*C_v(\theta_{sat}-\theta)]\right\}}{\partial t} = -\frac{\partial}{\partial z}(q_i)-C_i^lS_p \tag{19}$$

$$q_i = C_i^lq_l - D_{i,s}^l\frac{\partial C_i^l}{\partial z} - D_{i,s}^v\frac{\partial\left(C_v\alpha_i^*C_i^l\right)}{\partial z} \tag{20}$$

### 2.1.3 Root water uptake

According to Li et al. (2001), root water uptake (sink term in Eq. 1 and 19) is modeled by:

$$S_p = \frac{\alpha_rF_iP_t}{\Delta z} \tag{21}$$



where $\alpha_r$ is the efficiency coefficient (between 0 and 1) and can be obtained from a prescribed stress function (Feddes et al.,

1978); $F_i$ is the fraction of root length density distribution; $\Delta z$ (m) is the thickness of the considered layer and $P_t$ (m s$^{-1}$) is the

potential transpiration rate, which may be obtained from the Penman-Monteith equation (Allen et al., 1999) and the leaf area

index.

The stress function considers physiological processes during root water uptake by employing four critical absolute values of

pressure head ($h_1$, $h_2$, $h_3$, $h_4$; Fig. 2), where $\alpha_r$ can be modeled based on the pressure head of soil water ($h_s$):

$$\alpha_r = \begin{cases} 0, & h_s < h_4 \text{ or } h_s > h_1 \\ \frac{h_s - h_4}{h_3 - h_4}, & h_4 \leq h_s \leq h_3 \\ \frac{h_s - h_1}{h_2 - h_1}, & h_2 \leq h_s \leq h_1 \\ 1, & h_3 \leq h_s \leq h_2 \end{cases} \tag{22}$$

Thus, these four critical values are important for the root water uptake simulation and can vary among different plant species.

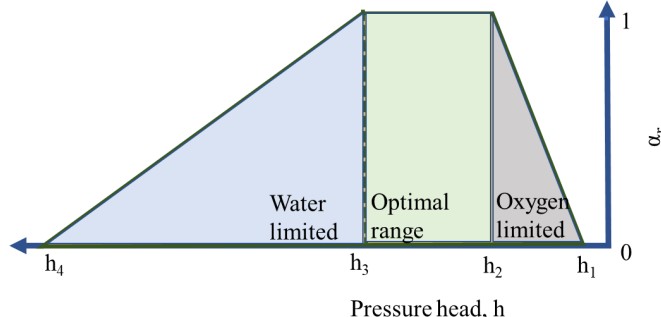

**Figure 2: Illustration of a stress function, based on Feddes et al. (1978).**

$F_i$ is defined as:

$$F_i = \frac{\int_z^{z+dz} R(x)\,dx}{\int_0^{Depth} R(x)\,dx} \tag{23}$$

where $z$ (m) is the depth of upper boundary of layer $i$; $z+dz$ is the depth of the lower boundary. *Depth* is the depth of the soil

column. $R(x)$ is a predefined root distribution function.

**2.1.4 Boundary conditions**

*Boundary conditions for soil water transport*

According to the Soil-Litter-Isotope model (Haverd and Cuntz, 2010), the coupled energy and moisture mass conservation

equations are solved at the soil-air interface:

$$\frac{1}{r_{bw}}(C_{vs} - C_{va}) = -D_v C_{vsat} \frac{\Delta h_r}{dz/2} - D_v h_r \frac{\Delta C_{vsat}}{dz/2} - K \frac{\Delta h}{dz/2} + K \tag{24}$$

$$R_{net} = \frac{C_p}{r_{bh}}(T_s - T_a) + \frac{\rho \lambda_E}{r_{bw}}(C_{vs} - C_{va}) - K_H \frac{\Delta T}{dz/2} \tag{25}$$





where $C_{vs}$ (m³ $_{\text{liquid water}}$ m⁻³$_{\text{air}}$) and $C_{va}$ (m³ $_{\text{liquid water}}$ m⁻³$_{\text{air}}$) are the concentrations of water vapor at the soil surface and atmosphere, respectively; $r_{bw}$ (s m⁻¹) and $r_{bh}$ (s m⁻¹) are the resistance to vapor and heat transfer at soil-air interface, respectively. $D_v$ (m² s⁻¹) is the diffusivity of vapor; $h_r$ is the relative humidity in soil. $K$ (m s⁻¹) and $K_H$ (W m⁻¹ K⁻¹) are the unsaturated hydraulic conductivity and thermal conductivity of the first layer. $R_{net}$ (W m⁻² K⁻¹) is the net radiation. $Cp$ (J m⁻³ K⁻¹) is the volumetric

heat capacity of air at constant pressure.

Eq. (25) and (26) are solved by a Jacobian iteration method at the beginning of each time step for soil surface temperature ($T_s$) and surface relative humidity ($h_{rs}$). These two unknowns can then be used to calculate surface evaporation flux; surface liquid and vapor fluxes; surface sensible heat flux; surface latent heat flux and heat flux into the soil (Haverd and Cuntz, 2010).


The lower boundary condition for soil water flow can be set to free drainage (zero gradient of soil water pressure head at lower boundary), seepage surface (constant soil water pressure head at lower boundary), or zero water flux. The lower boundary condition for heat transport can be zero heat flux or zero temperature gradient.

*Boundary conditions for isotope transport*

Isotopic evaporation flux through the soil surface to the atmosphere is calculated using the Craig-Gordon (1965) model:

$$E_i = \frac{\alpha_k}{r_{bw}}\left(C_{vs}C_{i,s}^l\alpha_i^*(T_s) - C_i^{va}\right) \tag{26}$$

where $C_i^{va}$ (kg m⁻³) is the isotopic concentration in atmosphere; $C_{i,s}^l$ (kg m⁻³) is the isotopic concentration at soil surface; $\alpha_k$ and $\alpha_i^*$ are the kinetic fractionation coefficient and equilibrium fractionation coefficients; $T_s$ (°C) is the soil surface

temperature; isotopic flux and $E_i$ (m s⁻¹) is the surface isotopic flux, which can be specified for soil:

$$E_i = -\frac{D_{i,1}^v}{\frac{dz}{2}}\left(C_{vs}C_{i,s}^l\alpha_i^*(T_s) - C_{v,1}C_{i,1}^l\alpha_i^*(T_1)\right) - \frac{D_{i,1}^l}{\frac{dz}{2}}\left(C_{i,s}^l - C_{i,1}^l\right) + q_{ls}C_{i,s}^l \tag{27}$$

Combining Eq. (26) and Eq. (27), the final expression for surface isotopic concentration is:

$$\frac{\alpha_k}{r_{bw}}\left(C_{vs}C_{i,s}^l\alpha_k(T_s) - C_i^{va}\right) = -\frac{D_{i,1}^v}{dz/2}\left(C_{vs}C_{i,s}^l\alpha_i^*(T_s) - C_{v,1}C_{i,1}^l\alpha_i^*(T_1)\right) - \frac{D_{i,1}^l}{dz/2}\left(C_{i,s}^l - C_{i,1}^l\right) + q_{ls}C_{i,s}^l \tag{28}$$

where $\alpha_k$ can be written as (Mathieu and Bariac, 1996; Haverd and Cuntz, 2010):

$$\alpha_k = \left(\alpha_{diff}\right)^{nk} \tag{29}$$

with

$$nk = \frac{(\theta_{sat}-\theta_r)n_a + (\theta_{sat}-\theta_s)n_s}{\theta_{sat}-\theta_r} \tag{30}$$

where $n_a$ and $n_s$ are 0.5 and 1.0, respectively. $\theta_r$ (m³ m⁻³) and $\theta_s$ (m³ m⁻³) are residual and soil surface water content, and $q_{ls}$ (m s⁻¹) is the liquid flux at the soil surface. The subscripts $s$ and $I$ represent the soil/air surface and the first layer of soil, respectively.

As the only unknown variable, isotopic concentration at the soil surface, $C_{i,s}^l$ (kg m⁻³), can be solved readily. More details can



be found in Haverd and Cuntz (2010).

The lower boundary conditions of isotope transport are determined by soil water flux or can be customized. Generally, the lower boundary condition for isotope transport is zero gradient or zero flux.

**2.1.5 Numerical implementations**

To solve Eq. (1), Eq. (5), and Eq. (20) simultaneously, a variable-step, variable-order (VSVO) Adams-Bashforth-Moulton method (Shampine, 2002) is adopted, which is integrated into the ODE solver (ode113) provided in MATLAB. It should be noted that ode113 is sufficient for a homogeneous soil column. When soil is layered with different hydraulic properties, the problem could become stiff, because the water flux can vary drastically at the layer interface. In such a case, ode23tb, which

is designed for stiff problems, may perform better.

**2.1.6 Modeling efficiency**

To evaluate the model performance quantitively, the Nash-Sutcliffe efficiency ($NSE$) (Nash and Sutcliffe, 1970) is used:

$$NSE = 1 - \frac{\sum_{t=1}^{T_{ol}} (M_0(t) - M_m)^2}{\sum_{t=1}^{T_{ol}} (M_0(t) - \overline{M_0})^2} \tag{31}$$

where $M_0$ and $M_m$ are observations (measurements) and modeling values (simulations) respectively. $\overline{M_0}$ is the average of

observations over time; $t$ is the temporal point and $T_{ol}$ is the total temporal points. $NSE$ ranges between negative infinity to 1. $NSE$ of 1 is indicative of excellent performance of the model in predicting the temporal variations of variables, while a $NSE$ of 0 suggests the model can only reflect average values. A negative $NSE$ implies poor performance of the model in regenerating the temporal variations of variables.

However, $NSE$ may be biased when the variables vary in a narrow range and is more likely to produce a large negative $NSE$ because the denominator of Eq. (31) becomes small. Thus, the coefficient of determination ($R^2$) is used to evaluate the goodness of simulations compared to observations:

$$R^2 = \left( \frac{\sum_{t=1}^{T_{ol}} (M_0(t) - \overline{M_0})(M_m(t) - \overline{M_m})}{\sqrt{\sum_{t=1}^{T_{ol}} (M_0(t) - \overline{M_0})^2 (M_m(t) - \overline{M_m})^2}} \right)^2 \tag{32}$$

where $\overline{M_m}$ is the average of modeling values.


Although root mean square error ($RMSE$) is widely used to evaluate model performance, we decided to use mean absolute error ($MAE$) to illustrate the model is not biased because the residuals between simulations and measurements are non-normally distributed (Sprenger et al., 2018; Chai and Draxler, 2014). Furthermore, $MAE$ reflects the realistic errors from modeling while $RMSE$ tends to increase the error through the calculation. $MAE$ is calculated by:





$MAE = \frac{1}{T_{ol}} \sum_{t=1}^{T_{ol}} |M_0(t) - M_m(t)|$                                                    (33)

### 2.2 Site descriptions

### 2.2.1 École Polytechnique Fédérale de Lausanne (EPFL), Switzerland

Data were collected by Nehemy et al. (2021) from a vegetated (*Salix. viminalis*) continuously weighed soil lysimeter with a 2.5 m depth and 1.12 m$^2$ basal area (data are available at https://zenodo.org/record/4037240#.Y029l3bMKUk). The study was

performed at the École Polytechnique Fédérale de Lausanne (EPFL), Switzerland (46°31''N, 6°33''E). The experiments were conducted over the course of 50 days from 10 May 2018 to 29 June 2018. During the experiment, the air temperature, relative humidity, and precipitation were automatically recorded at 15-minute intervals by a weather station located 5 m from the lysimeter. The daily mean air temperature increased over the study period, with daily minimum and maximum values varying between 10 °C and 30 °C (Fig. 3a). Conversely, the atmospheric relative humidity showed a decreasing trend and varied

between 0.25 and 0.99 (Fig. 3b). In addition to precipitation water input, the lysimeter was occasionally irrigated with tap water during the experiment (Fig. 3c). In total, the lysimeter received 199 mm from rainfall and 489 mm from irrigation. Most precipitation and irrigation events occurred in the first 35 days, while there were only irrigation events in the final 15 days. Water from each rainfall and irrigation event was sampled and analyzed for isotopic compositions.

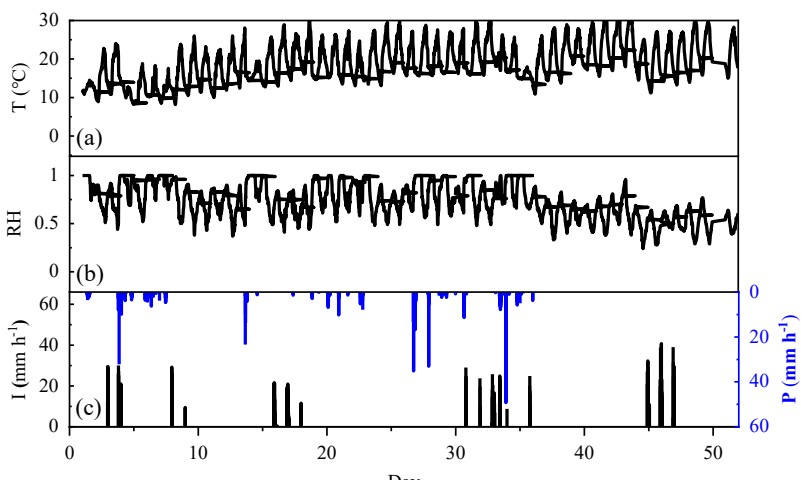

**Figure 3: Air temperature, relative humidity, precipitation, and irrigation events at EPFL in Switzerland: (a) air temperature, (b) relative humidity, and (c) precipitation events (P, blue bar) and irrigation events (I, black bar). Data from Nehemy et al. (2021).**

Two small basket willows (*Salix. viminalis*) were planted due to their drought-resistant capability. The sap flow, leaf water potential at the crown, and stem radius were recorded at 15-minute intervals. The roots of the trees extended to 2 m depth with the greatest root length density between 0 m and 0.5 m (Nehemy et al., 2021).


Frequency domain reflectometry probes were used to monitor and record volumetric soil water content at 15-minute interval


from 4 depths (0.10, 0.25, 1.25, and 1.75 m), where each measurement consisted of 2 replications. Two replicate bulk soil samples for soil water extraction were collected every four days at multiple depths (0.1, 0.5, 0.8, and 1.5 m). Soils from 0 to 0.1 m depth were sampled vertically, and the remaining samples were collected horizontally (Nehemy et al., 2021).


The isotopic signals of soil water and precipitation were measured independently at the Central Environmental Laboratory at EPFL, and the Hillslope Hydrology Laboratory at the University of Saskatchewan (Benettin et al., 2021). Detailed information of the lysimeter, plants, soils, sampling, and extraction methods can be found in Nehemy et al. (2021).

**2.2.2 HBLFA Raumberg-Gumpenstein, Austria**

Precipitation and seepage water from lysimeters were collected from May 2002 to February 2007 by Stumpp et al. (2012) at the HBLFA Raumberg-Gumpenstein, Austria (data are available at https://www.pc-progress.com/en/Default.aspx?h1d-lib-isotope). During the experiment, the air temperature exhibited a sinuous variation, ranging between -15 °C and 27 °C (Fig. 4a), with a mean of 8.2 °C. Similarly, the atmospheric relative humidity showed seasonal fluctuations and varied between 0.30 and 0.99 (Fig. 4b), with a mean of 0.89. In addition, most precipitation events occurred during the summer (Fig. 4c), with a daily

mean rainfall of 2.8 mm day$^{-1}$.

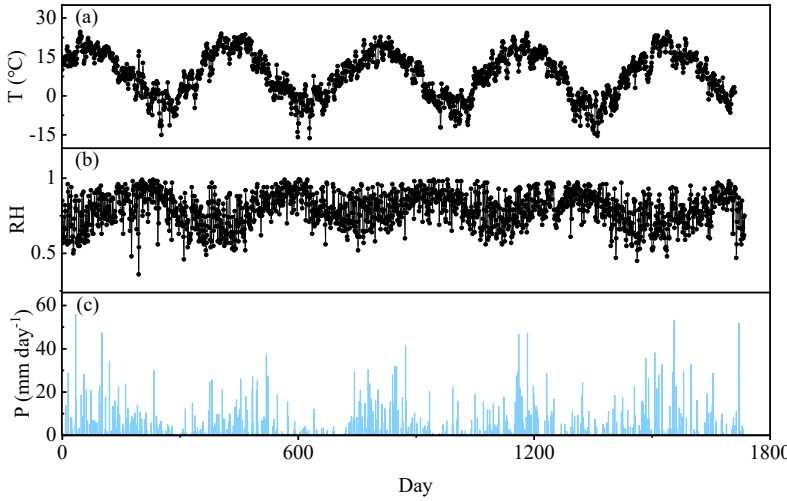

**Figure 4: Air temperature, relative humidity, and precipitation events at HBLFA Raumberg-Gumpenstein in Austria: (a) air temperature (T), (b) air relative humidity (RH), and (c) daily precipitation amount (P). Data from Stumpp et al. (2012)**

Five lysimeters were used to investigate the influence of land cover and fertilization on soil water and solute transport by

Stumpp et al. (2012). For simplicity, the current study used only lysimeter-3 for the comparison of numerical simulations. Lysimeter-3 had a surface area of 1 m$^2$, depth of 1.5 m, and consisted of three soil horizons (0 - 0.25 m, 0.26 - 1.0 m, 1.0 - 2.0 m). The lysimeter and was filled with three horizons of undisturbed Dystric Cambisol (Stumpp et al., 2012). A fluvioglacial sediments layer (0.05 m in thickness) was placed at the bottom of the soil profile as the lower boundary.



Each year the lysimeter was planted with winter rye, which has a maximum rooting depth of 1 m. Weekly precipitation and drainage water samples from the bottom of Lysimeter-3 were collected between May 2002 and February 2007. Isotopic compositions were analyzed by using dual-inlet mass spectrometry. Further information about the site and experimental procedures can be obtained from Stumpp et al. (2012).

### 2.3 Model validation

To ensure model accuracy, we conducted both theoretical and semi-analytical tests, followed by validation of the model under field conditions.

### 2.3.1 Theoretical tests

The six theoretical tests performed in this study were initially designed by Mathieu and Bariac (1996), have been widely used to validate the accuracy and stability of isotope transport models (Braud et al., 2005; Haverd and Cuntz, 2010; Zhou et al.,
2021). In the simulations, water in the 1 m soil column (Yolo light clay) can only escape through evaporation from the top of the column. The simulations are conducted for 250 days, using the specified parameters and values (Table. 1 and 2).

**Table 1: Parameters of the six theoretical tests**

| Test | $\alpha$ | $D_i^v$ | $D_i^l$ | $\alpha_k$ | $\delta_a$ |
|------|----------|---------|---------|------------|------------|
| 1 | 1 | $D_v$ | 0 | 1 | $\delta_{soil\ water}$ |
| 2 | 1 | $D_v$ | 0 | 1 | $\delta_a$ |
| 3 | $\alpha$ | $D_v$ | 0 | 1 | $\delta_{soil\ water}$ |
| 4 | $\alpha$ | $D_v$ | 0 | 1 | $\delta_a$ |
| 5 | $\alpha$ | $D_v$ | $D_i^l$ | 1 | $\delta_a$ |
| 6 | $\alpha$ | $D_i^v$ | $D_i^l$ | $\alpha_k$ | $\delta_a$ |

$\alpha$ is the equilibrium fractionation coefficient.
$D_i^v$ is the vapor diffusivity of isotopic species.
$D_i^l$ is the liquid diffusivity of isotopic species.
$\alpha_k$ is the kinetic fractionation coefficient.
$\delta_a$ is the atmospheric isotopic compositions: -120‰ and -15‰ for $\delta^2 H$ and $\delta^{18}O$, respectively.

The relationships between soil water content, pressure head, and unsaturated hydraulic conductivity are described by the Brooks-Corey (1964) model:

$$S=\frac{\theta-\theta_{res}}{\theta_{sat}-\theta_{res}}=\begin{cases}\left(\frac{h}{h_e}\right)^{-\lambda}, h \leq he \\ 1 \quad h \geq he\end{cases} \tag{34}$$

$$S^{\eta} = \begin{cases}\frac{k}{k_{sat}}, h \leq he \\ 1 \quad h \geq he\end{cases} \tag{35}$$

where $S$ is the effective saturation; $\theta$ (m³ m⁻³), $\theta_{sat}$ (m³ m⁻³) and $\theta_{res}$ (m³ m⁻³) are the actual, saturated, and residual soil water



contents; $h$ (m) is the pressure head; $h_e$ (m) is the air-entry value; $\lambda$ and $\eta$ are the shape coefficients, where $\eta = 2/\lambda + 3$ (Table 2).

**Table 2: Hydraulic properties of the simulated soil used in the theoretical tests.**

| $h_e$ (m) | $\lambda$ | $k_{sat}$ (m s$^{-1}$) | $\theta_{sat}$ (m$^3$ m$^{-3}$) | $\theta_{res}$ (m$^3$ m$^{-3}$) |
|---|---|---|---|---|
| -0.193 | 0.22 | $1.23\times10^{-7}$ | 0.35 | 0.01 |

The soil column is initially saturated and evaporates at a potential evaporation rate of $2\times10^{-7}$ m s$^{-1}$. Air temperature and relative humidity during the simulation period remain at 30°C and 0.2, respectively. Upper boundary conditions for soil water and isotope transport are calculated by Eq. (24), Eq. (25), and Eq. (28). Lower boundary conditions for soil water and isotope transport are zero fluxes.


For model set up, the column is divided vertically into 100 layers with a layer thickness of 0.01 m. The initial temporal step is set to 100 s and is self-adaptive up to a maximum of 500 s.

**2.3.2 Semi-analytical tests**

*Saturated and isothermal conditions*

An analytical solution of isotopic distribution in a saturated, isothermal soil column for steady state evaporation, is presented by Barnes and Allison (1983) as:

$$\delta_l^i = \delta_{l,sup}^i + (\delta_s^i - \delta_{l,sup}^i)e^{\frac{\rho q_{evap}}{D_{i,s}^l}z} \tag{36}$$

where $z$ (m) is the soil depth; $\delta_l^i$ (‰) is $\delta^2$H or $\delta^{18}$O at depth $z$; $\delta_{l,sup}^i$ (‰) is the $\delta^2$H or $\delta^{18}$O of supplied water from the bottom of the soil column; $\delta_s^i$ (‰) is the $\delta^2$H or $\delta^{18}$O at the soil surface; $\rho$ (kg m$^{-3}$) is the water density; $q_{evap}$ (m s$^{-1}$) is the

evaporation flux and; $D_{i,s}^l$ (m$^2$ s$^{-1}$) is the isotopic diffusion coefficient of isotopic species $i$ ($^2$H or $^{18}$O) in soil.

The soil properties and initial soil water contents in the semi-analytical tests were identical to those used in the theoretical tests. However, the potential evaporation rate from the upper boundary was assumed to be $1\times10^{-8}$ m s$^{-1}$ in the semi-analytical tests, whereas the theoretical test used a value of $2\times10^{-7}$ m s$^{-1}$. The initial $\delta^2$H and $\delta^{18}$O were set to 0‰. The lower boundary condition

in this simulation was switched to upward water flux, which is equivalent to the evaporation rate. The lower boundary condition of isotope transport was constant concentration, and equivalent to the initial isotopic compositions of soil water.

*Unsaturated and non-isothermal conditions*

Barnes and Allison (1984) developed a semi-analytical solution to predict $\delta^2$H and $\delta^{18}$O profiles under unsaturated, non-



isothermal conditions, which can be solved implicitly:

$$\frac{d\delta_i}{dz} + \frac{\delta_i - \delta_{i,sup}}{z_l + h_r z_v} = \frac{h_r z_v (\alpha_k - \alpha_*)}{z_l + h_r z_v} \frac{d\left[ln\left(h_r \rho C_{v,sat}(\alpha_k - \alpha_*)\right)\right]}{dz} \tag{37}$$

$$z_l = \frac{D_{i,s}^l}{q_{evap}} \tag{38}$$

$$z_v = \frac{D_{i,s}^v C_{v,sat}}{q_{evap}} \tag{39}$$

where $\delta_i$ (‰) is the $\delta^2H$ or $\delta^{18}O$ at the $i^{th}$ layer; $z$ (m) is the depth; $\delta_{i,sup}$ (‰) is the $\delta^2H$ or $\delta^{18}O$ of the bottom supplement,

which is the water supplied to the bottom of the soil column during the simulation; $z_l$ (m) and $z_v$ (m) are the liquid and vapor

characteristic lengths; $h_r$ is the relative humidity within the soil; $\alpha_k$ and $\alpha_*$ are the kinetic and equilibrium fractionation

coefficients; $C_{v,sat}$ (m$^3$ $_{liquid\ water}$ m$^{-3}$ $_{air}$) is the saturated vapor concentration and $q_{evap}$ (m s$^{-1}$) is the evaporation flux.

Soil configuration, initial isotopic compositions, and upper boundary conditions were identical to the theoretical Test 6, but

heat transport was considered with a constant net radiation of 200 W m$^{-2}$, whereas the theoretical tests used a value of 0 W m$^{-2}$. The initial soil water content was set at 70% of its saturated value, and the rate of water supplement from the bottom of the

profile was equal to the evaporation rate at each time step (Haverd and Cuntz, 2010).

**2.3.3 A short-term simulation at EPFL**

The experimental soil from EPFL consisted of 50% local loamy sand and 50% lacustrine sand. The initial value of saturated

hydraulic conductivity of the experimental soil was set to 1×10$^{-5}$ m s$^{-1}$, which is in the same magnitude as the values from

HYDRUS-1D (4×10$^{-5}$ m s$^{-1}$ for loamy sand and 7×10$^{-5}$ m s$^{-1}$ for sand) (Šimůnek et al., 2013).

Soil water retention curves were determined by fitting the relationship between measured soil water potential and soil water

content data using the Brooks-Corey model (1964) (Eq. 34 and 35). The fitted $\theta_{sat}$, $\theta_{res}$, $h_e$, and $\lambda$ of each horizon were used as

initial values. The parameters can then be optimized by minimizing the objective function:

$$y_i = \sum_{i=1}^{N_{layer}} (\theta_0 - \theta_m)^2 \tag{40}$$

where $\theta_0$ is the measured soil water content, and $\theta_m$ is the modeled soil water content. The dispersivity length of deuterium

was adopted from Stumpp et al. (2012) and adjusted according to the model efficiency (Eq. 31 - 33).

Continuous initial soil water content and $\delta^2H$ profiles were obtained through linear interpolation between measurements at

different depths within the 2 m depth profile. Due to the absence of soil temperature measurements, the initial soil temperature

was assumed to be uniformly distributed throughout the column and equal to air temperature.



The upper boundary conditions of soil water, heat, and isotope transport were determined by Eq. (24), Eq. (25), and Eq. (28),

while the lower boundary conditions for water and heat transport were defined as seepage surface and zero temperature gradient, respectively. The lower boundary condition was set to zero gradient for isotope transport, which means convection was the only component of isotopic flux across the lower boundary of the soil profile and there were no isotopic sources below the lower boundary. The simulations used $\delta^2H$ because more measurements of $\delta^2H$ were available. Nevertheless, using $\delta^{18}O$ would result in similar conclusions.


The distribution of fine roots (< 2 mm diameter) were sampled at the end of the experimental period and the fine root length density was described throughout the profile by a 5-degree polynomial function. The root length density pattern was assumed to be constant during the experimental period. We set the water stress function parameters $h_1$, $h_2$, $h_3$, and $h_4$ according to the values obtained from HYDRUS-1D (Šimůnek et al., 2013), which are 0.03, 0.205, 8, and 20 m respectively.


The simulation was performed over a period of 50 days using 200 soil layers and a spatial step of 0.01 m. The temporal step size was self-adaptive, with an initial temporal step of 100 s. Meteorological parameters such as air temperature, air relative humidity, and precipitation were assumed to be constant within the 15-minute measurement intervals.

### 2.3.4 A long-term simulation at HBLFA Raumber-Gumpenstein

The solute transport and soil hydraulic parameter values optimized by Stumpp et al. (2012) were used in our simulation. The soil hydraulic properties were described by the van Genuchten (1980) model. Because the measured saturated hydraulic conductivities vary greatly within different soil horizons (Stumpp et al., 2012), water flux at the interface of different soil layers may vary drastically. Therefore, to minimize the oscillation of numerical solutions, the ode23tb solver, which is designed for stiff problems (MathWorks, 2022), was used in this simulation.


The initial water content and $\delta^{18}O$ profiles were provided by Stumpp et al. (2012). The upper boundaries of soil water and heat transport were calculated by Eq. (24) and Eq. (25), while the upper boundary of isotope transport was calculated by Eq. (28). The lower boundary condition for soil water flow was defined as seepage surface. Zero temperature and zero isotopic concentration gradients were set as the lower boundary conditions of heat and isotope transport, respectively.


The root distribution varied during the growing season at this site and has been described in HYDRUS-1D (Stumpp et al., 2012), where the water stress is described by the Feddes model (1978). Following this approach, the water stress function parameters $h_1$, $h_2$, $h_3$, and $h_4$ were set as 0, 0.01, 5, and 160 m, respectively.

Environmental parameters such as air temperature, air relative humidity, and precipitation were assumed to be constant within

each hour. The simulation length was 1736 days and initial temporal step was 1 day. The soil column was divided into 200

layers with a spatial step of 0.01 m in our model.

### 3 Results

#### 3.1 Validation by theoretical tests

The MOIST model was evaluated using 6 theoretical tests. The results indicate volumetric soil water content at the soil surface

was close to the residual soil water content (Fig. 5a), suggesting that a drying layer appeared at the top 0.005 m after a 250-

day evaporation period. In the drying layer (Fig. 5b), where the soil water flux was dominated by vapor transport, and the

liquid flux was nearly zero. However, liquid flux dominated below the drying layer (Fig. 5b).

Test 1 was designed to test the model stability. There was no isotopic gradient between soil surface and atmosphere and, thus,

there was no fractionation (Table 1). As expected, the soil isotope profile generated by the model is identical to the initial

profile (Figs. 5c and 5d).

Test 2 had identical parameters to Test 1, except that the atmospheric isotopic compositions were more depleted than that of

soil water. The isotopic compositions at the soil surface should be close to atmospheric isotopic ratios within the drying layer

after a long-term evaporation period. This test also checks the accuracy of the simulated upper boundary conditions. As

expected, the simulated $\delta^2$H and $\delta^{18}$O at the soil surface became similar to the atmospheric isotopic compositions (Figure 5c

and 5d). This similarity was due to the diffusion of lighter water molecules through the vapor phase from the atmosphere to

the soil water in the absence of fractionation. The soil water isotopic compositions below the drying layer were the same as

those in Test1.

Test 3 had the same parameters as Test 1, except that equilibrium fractionation was considered. The simulated results show

enriched $\delta^2$H and $\delta^{18}$O at the soil surface (Figs 5c and 5d). The enrichment was expected because lighter water molecules were

preferentially evaporated, leaving heavier water molecules in the liquid phase, and resulting in an enrichment of $\delta^2$H and $\delta^{18}$O

in the remaining surface soil water.

Test 4 was similar to Test 3 but used an atmospheric isotopic signal that was more depleted (Table 1). The results of Test 4

showed that $\delta^2$H and $\delta^{18}$O were enriched in the surface soil, albeit not to the same extent as that of Test 3 (Figs. 5c and 5d). In

Test 4, the back diffusion processes from air to soil, as described in Test 2, resulted in isotopic depletion of the surface soil as





compared to Test 3.

Test 5 had identical parameters to Test 4 except that the liquid diffusion coefficient was not zero. The peak $\delta^2H$ and $\delta^{18}O$ from

Test 5 were slightly lower than that from Test 4 but extended deeper (Figs. 5c and 5d). This is because the liquid diffusion is

considered and isotopic species were diffused downward due to concentration gradient, resulting in a deeper extension and

smaller enrichment peaks as compared to Test 4.

The parameters for test 6 were all set to realistic values and included kinetic fractionation. As expected, the isotope peak values

from Test 6 were greater in magnitude and extended to a greater depth than that of the other tests (Figs. 5c and 5d). The

magnitude was greater because the kinetic fractionation and equilibrium fractionation processes resulted in more highly

enriched profiles.

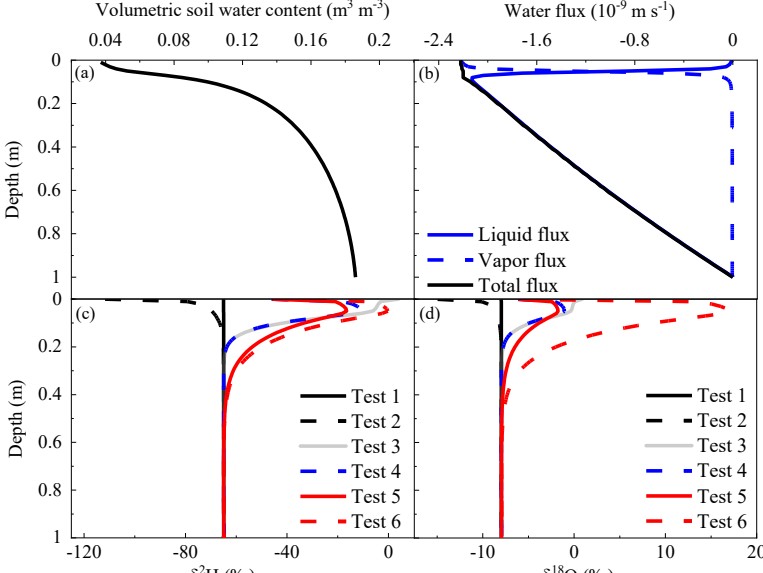

**Figure 5: Results from the theoretical tests performed at the end of the simulation. (a) Volumetric soil water content; (b) liquid and vapor flux; (c) soil $\delta^2H$ curves and (d) soil $\delta^{18}O$.**

**3.2 Validation by semi-analytical tests**

**3.2.1 Saturated and isothermal conditions**

The results of the semi-analytical test with saturated and isothermal conditions indicated that the soil was saturated everywhere,

except in the near surface of the profile where there was a slight decrease in soil water content (Fig. 6a). When the water supply

rate at the bottom of soil column is equal to the evaporation rate, steady state evaporation is obtained. The liquid flux was

uniform with depth and equal to the evaporation rate ($1 \times 10^{-8}$ m s$^{-1}$), while the vapor flux was nearly zero (Fig. 6b).


Both the simulated $\delta^2H$ and $\delta^{18}O$ were in good agreement with the analytical solutions (Figs. 6c and 6d). Due to evaporation

from the soil surface, isotopic fractionation caused the lighter isotope molecules to evaporate first, which in turn lead to the

isotopic enrichment in the surface soil. Because the bottom of the soil profile was continuously supplied with water throughout

the simulation, water that was evaporated at the soil surface was replenished with new water from below. As a result of this

steady supply of water, a drying layer did not form during evaporation. Furthermore, because the supplied water was consistent

in isotopic composition throughout the simulation, the final isotope concentrations were exponentially distributed with soil

depth, with relatively enriched water at the soil surface and depleted water at depth.

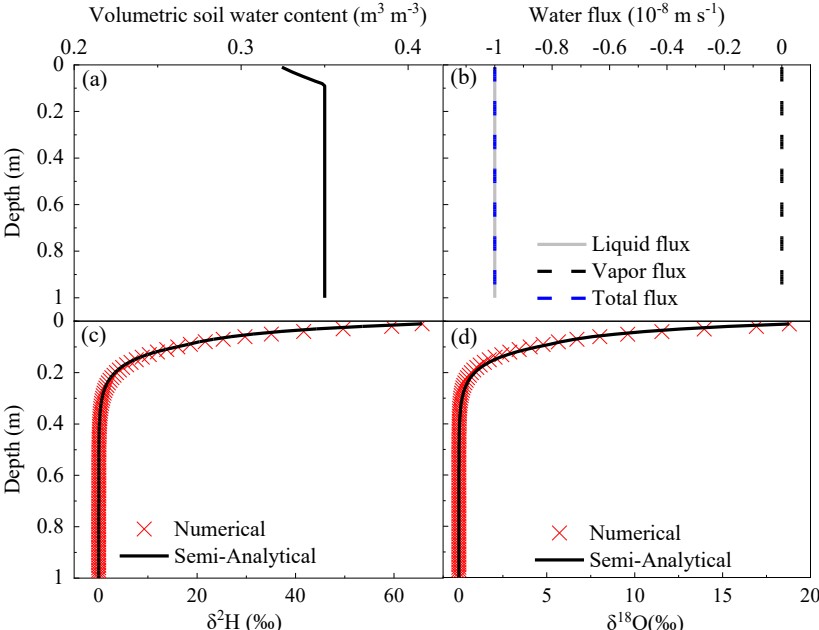

**Figure 6: Results of the semi-analytical test after steady state evaporation was reached under saturated and isothermal conditions.**

**(a) Volumetric soil water content; (b) liquid and vapor flux profiles; (c) $\delta^2H$ results of numerical and semi-analytical solutions; (d)**
        **$\delta^{18}O$ results of numerical and semi-analytical solutions.**

### 3.2.2 Unsaturated and non-isothermal conditions

Under unsaturated conditions, a drying layer appeared at the soil surface (Fig. 7a) and water flow in this region was dominated

by vapor diffusion (Fig. 7b). However, because steady state was achieved, the total water flux within the column was uniform

with depth (Fig. 7b).

The $\delta^2H$ and $\delta^{18}O$ increased sharply with depth until reaching their peak values at approximately 0.02 m depth, which is the

maximum depth of the drying layer (Figs. 7c and 7d). In contrast to the saturated conditions (Figs. 6c and 6d), the maximum



$\delta^2$H and $\delta^{18}$O appeared below the soil surface in the unsaturated system (Figs. 7c and 7d). This is because the soil water content

at the air-soil interface was close to residual soil water content in the unsaturated system: when a drying layer forms, the air

that invades into the drying layer results in a downward shift of the isotope peak values in the unsaturated system as compared

to the saturated system.

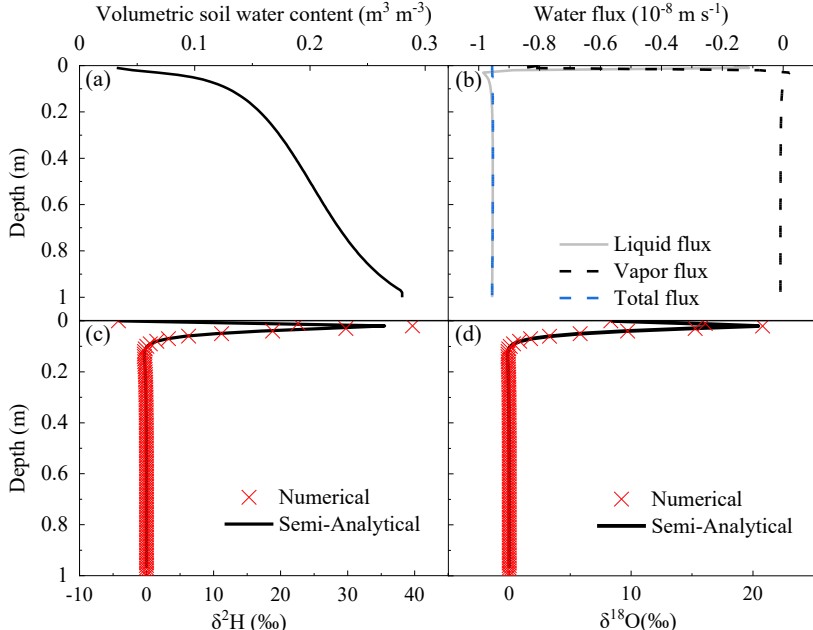

**Figure 7: Results of the semi-analytical test after steady state evaporation was reached under unsaturated and non-isothermal**

**conditions. (a) Volumetric soil water content; (b) liquid and vapor flux; (c) $\delta^2$H results of numerical and semi-analytical solutions;**

**(d) $\delta^{18}$O results of numerical and semi-analytical solutions.**

**3.3 Validation by a short-term experiment at EPFL**

**3.3.1 Soil properties**

Although the soil was packed uniformly in the lysimeter, there was still considerable heterogeneity as indicated by the different

soil water retention curves from different depths (Fig. 8). The fitted water retention functions at 0.25 and 0.75 m were almost

identical (Figs. 8a and 8b), with a mean $h_e$ of -0.031 and $\lambda$ of 0.135 (Table 3). These values are used to describe soil hydraulic

properties within the top 1 meter. However, the retention curves at depths of 1.25 and 1.75 m differed, where $h_e$ was -0.046

and -0.025, and $\lambda$ was 0.164 and 0.120, respectively (Table 3). We ignored hysteresis because considering hysteresis does not

typically improve the solute concentration simulations (Mitchell and Mayer, 1998; Pickens and Gillham, 1980), which is

consistent with Šimůnek et al. (2013) who showed that the non-hysteretic assumption is acceptable under most scenarios.

Based on the difference in water retention curves at different depth (Fig. 8), we divided the soil column, in our simulation, into

three horizons: 0-1.0 m, 1.0-1.5 m, and 1.5-2.0 m (Table 3).



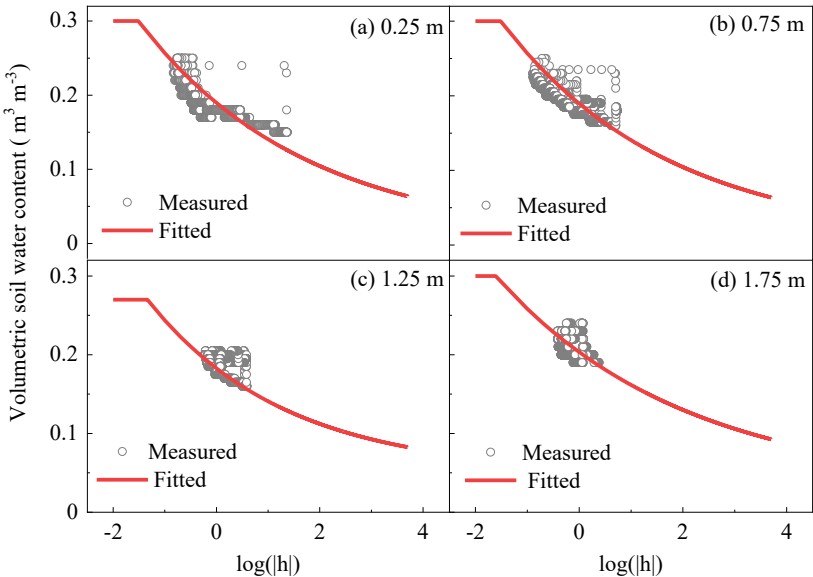

**Figure 8: Measured and fitted water retention functions at soil depths of (a) 0.25 m, (b) 0.75 m, (c) 1.25 m, and (d) 1.75 m from the experiment at EPFL in Switzerland.**

Table 3: Soil hydraulic parameters used in the short-term simulation at EPFL in Switzerland.

| | $h_e$ (m) | $\lambda$ | $k_{sat}$ (m s$^{-1}$) | $\theta_{sat}$ (m$^3$ m$^{-3}$) | $\theta_{res}$ (m$^3$ m$^{-3}$) |
|---|---|---|---|---|---|
| Horizon 1 (0 – 1.0 m) | -0.031 | 0.135 | 3.50×10$^{-5}$ | 0.30 | 0.01 |
| Horizon 2 (1.0 - 1.5 m) | -0.046 | 0.164 | 5.25×10$^{-5}$ | 0.27 | 0.05 |
| Horizon 3 (1.5 - 2 m) | -0.025 | 0.120 | 3.50×10$^{-5}$ | 0.30 | 0.03 |

### 3.3.2 Temporal variation of soil water content

The simulated temporal variations of soil water content agreed with the measurements at 0.25, 0.50, and 1.25 m (Fig. 9), with *NSE* values of 0.65, 0.69, and 0.67, and $R^2$ of 0.89, 0.84, and 0.89, respectively (Table 4). At 1.75 m, *NSE* was small (0.01) between the 30th and 33rd day, but the respective $R^2$ was moderate (0.53), and the soil water content was underestimated by 0.02 m$^3$ m$^{-3}$ on average (Fig. 9). The *MAE* values throughout the profile were reasonable with the values of 0.01, 0.01, 0.02, and 0.02 m$^3$ m$^{-3}$ at 0.25, 0.50, 1.25, and 1.75 m depths, respectively. In summary, our model had a good performance regenerating temporal soil water distributions with an overall *NSE* of 0.66, $R^2$ of 0.82, and *MAE* of 0.02 m$^3$ m$^{-3}$ (Table 4).





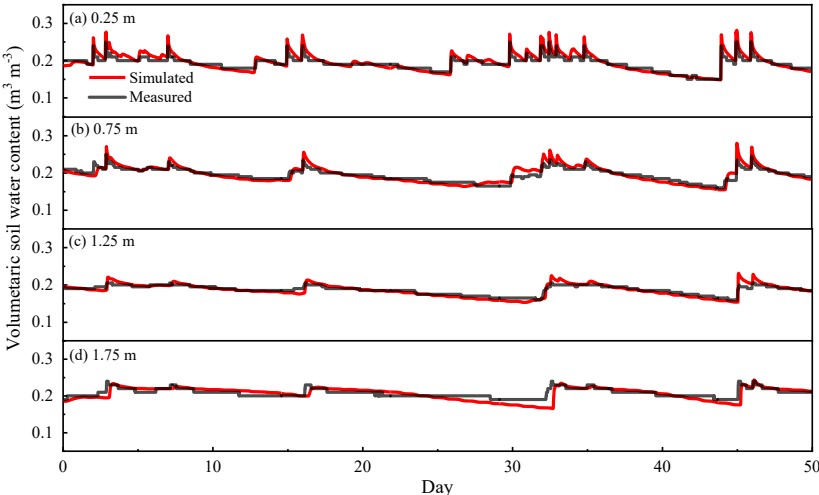

**Figure 9:** Temporal variation of measured and simulated soil water contents at 0.25, 0.5, 1.25, and 1.75 m depths during the 50-day study period at EPFL in Switzerland.

**Table 4: Nash-Sutcliff efficiencies (*NSE*), coefficient of determination (*R²*), and mean absolute error (*MAE*) determined by comparing measured and simulated soil water contents from the simulation at EPFL in Switzerland.**

|  | 0.25 m | 0.75 m | 1.25 m | 1.75 m | Overall |
|---|---|---|---|---|---|
| *NSE* | 0.65 | 0.69 | 0.67 | 0.01 | 0.66 |
| *R²* | 0.89 | 0.84 | 0.89 | 0.53 | 0.82 |
| *MAE* ($m^3\ m^{-3}$) | 0.01 | 0.01 | 0.02 | 0.02 | 0.02 |

### 3.3.3 Temporal variation of $\delta^2H$ in soil

Our model also had an acceptable performance in simulating isotope transport within soil, with an overall *NSE* of 0.69, *R²* of 0.70, and *MAE* of 11.84‰ (Table 5). The simulated $\delta^2H$ followed most of the measured data (Fig. 10). The water spike ($\delta^2H=255.56‰$), irrigated on the 7th day, was detected by both measurements and simulations at 0.1 m depth. The high *NSE* and *R²* values at 0.1 m (*NSE* = 0.83; *R²* = 0.86) and 1.5 m (*NSE* = 0.52, *R²* = 0.71) (Table 5), suggest excellent model performance. However, the *MAE* at 0.1 m depth was the largest (22.74‰) because the measured peak value was delayed approximately 4 days in comparison to the simulated value.

The *NSE* at 0.5 and 0.8 m depths were negative and close to zero, suggesting overestimations from the simulations (Fig. 10b and 10c), which is also evidenced by the relatively high *MAE* values of 11.80‰ and 7.89‰ at the two depths, respectively (Table 5). However, *R²* was 0.63 at both depths (Table 5), indicating that the model can reflect the temporal trend at 0.5 and 0.8 m depths.



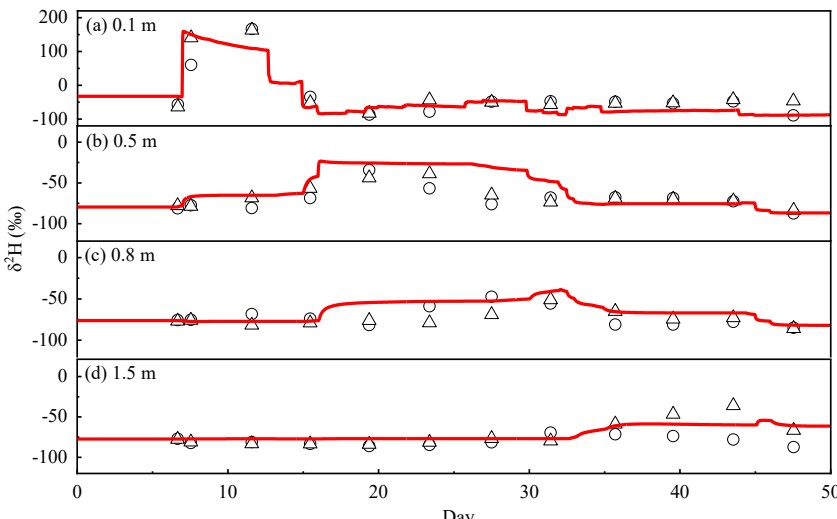

**Figure 10: Temporal variation of δ²H at 0.1, 0.5, 0.8, and 1.5 m depths for the experiment at EPFL in Switzerland. Circles and triangles represent field measurements (2 replications). Red lines are simulated values.**

**Table 5: Nash-Sutcliff efficiencies (*NSE*), coefficient of determination (*R²*), and mean absolute error (*MAE*) determined by comparing measured and simulated soil water δ²H from the simulation at EPFL in Switzerland.**

|            | 0.10 m | 0.50 m | 0.80 m | 1.50 m | Overall |
|------------|--------|--------|--------|--------|---------|
| *NSE*      | 0.83   | -0.76  | -0.49  | 0.52   | 0.69    |
| *R²*       | 0.86   | 0.63   | 0.63   | 0.71   | 0.70    |
| *MAE (‰)*  | 22.74  | 11.80  | 7.89   | 4.92   | 11.84   |

**3.4 Validation by a long-term experiment at HBLFA Raumber-Gumpenstein**

The MOIST model, as well as the revised HYDRUS-1D (rHS) by Stumpp et al. (2012), reproduced temporal variations of δ¹⁸O of drainage water (Fig. 11). Compared to the simulated results from rHS, MOIST had a better ability in predicting the

measured values. These measurements were also used to evaluate the recently revised HYDRUS-1D (rHZ) by Zhou et al. (2021). The simulations from rHZ are not shown here, but the statistics of the model performances are included (Table 6).

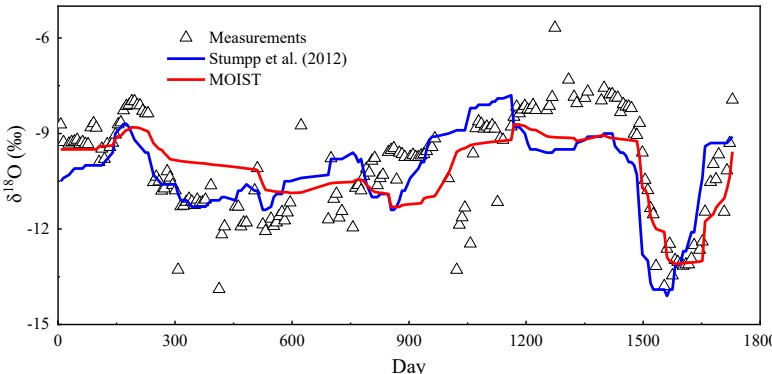

**Figure 11. δ¹⁸O from seepage water over the course of the experiment at HBLFA Raumberg-Gumpenstein in Austria. Included are the measurement values as well as the modeling results from MOIST and that of Stumpp et al. (2012).**



The *NSE* of MOIST was 0.47, which is greater than that of the rHS (0.31) and rHZ (0.19). The $R^2$ of MOIST was 0.49, which

is also greater than rHS (0.40) and rHZ (0.30). Moreover, the *MAE* of MOIST was 0.92‰, which is slightly smaller than the

1.00‰ from rHS. All these indices illustrate that MOIST outperformed rHS and rHZ over this long-term simulation period.

**Table 6. Nash-Sutcliffe efficiency coefficients (*NSE*), coefficient of determination (*$R^2$*), and mean absolute error (*MAE*) of $\delta^{18}O$ for models used in the simulations at HBLFA Raumberg-Gumpenstein in Austria.**

|  | MOIST | Revised HYDRUS-1D (Stumpp et al., 2012) | Revised HYDRUS-1D (Zhou et al., 2021) (Based on Craig-Gordon model) |
|---|---|---|---|
| *NSE* | 0.47 | 0.31 | 0.19 |
| *$R^2$* | 0.49 | 0.40 | 0.30 |
| *MAE* (‰) | 0.92 | 1.00 | *N/A* |

**4 Discussion**

**4.1 Model stability using thicker soil layers**

The MOIST model successfully passed the six theoretical tests, confirming its accuracy and stability. Note that Test 1 is

specifically designed to test the model stability. Normally, the finer the spatial discretization of the model, the better

performance and the more time is required for the model to converge on an acceptable result. The existing models, SiSPAT

(Braud et al., 2005), Soil-Litter-Iso (Haverd and Cuntz, 2010), and revised HYDRUS-1D (Zhou et al., 2021) used subsurface

soil layer thicknesses ranging from 0.001 to 0.01 m in the theoretical tests. However, even with a thickness of 0.1 m, simulated

$\delta^2H$, $\delta^{18}O$, soil water content, and soil water flux profiles estimated by MOIST remain accurate (Figure 12).

As mentioned above, available models in the literature obtain isotopic solutions by computing soil water and heat transport

first, and then solving isotope transport equations second, which means equations are solved with a segregated method. The

segregated method greatly relies on grid size (spatial steps). Interestingly, Braud et al. (2005) also mentioned that the stability

and accuracy of isotope transportation solutions are greatly affected by the thickness of subsurface layer. However, the coupled

method links all variables and equations implicitly. Compared to segregated methods, isotope results from coupled methods

are more stable and accurate (Pimenta and Alves, 2019; Pascau et al., 1996). A simplified example as described below

mathematically illustrates the difference between the segregated and coupled methods.

Assuming soil water and isotope transport can be described by following 'hypothetical' equations:

$$\frac{\partial \theta}{\partial t} = \frac{\partial q}{\partial z} \tag{41}$$

$$\frac{\partial (c\theta)}{\partial t} = 2\frac{\partial (cq)}{\partial z} \tag{42}$$

where $\theta$ (m³ m⁻³) is soil water content; $q$ (m s⁻¹) is soil water flux; $t$ (s) is time; $z$ (m) is depth; $c$ (kg m⁻³) is isotope concentration.





The unknowns, $\theta_z^{t+1}$ and $c_z^{t+1}$ (at the time point $t+1$) can be solved by the finite difference method (FDM) in each soil layer. When segregated method is used, $\theta_z^{t+1}$ can be calculated from Eq. (41), but an error term ($Err_1$) should be considered when rewriting Eq. (41) in a difference form:

$$\frac{\theta_z^{t+1} - \theta_z^t}{dt} + Err_1 = \frac{q_{z+1}^t - q_z^t}{dz} + Err_2 \tag{43}$$

where $dt$ (s) is the temporal step. Then, $\theta_z^{t+1}$ can be easily solved:

$$\theta_z^{t+1} = \left(\frac{q_{z+1}^t - q_z^t}{dz} + Err_2 - Err_1\right)dt + \theta_z^t \tag{44}$$

An extra error term is required to balance Eq. (41) according to Eq. (44):

$$\frac{\partial \theta}{\partial t} = \frac{\partial q}{\partial z} + Err_3 \tag{45}$$

where $Err_3$ contains $-Err_1$ and $Err_2$. Similarly, $c_z^{t+1}$ can be solved by Eq. (42) and Eq. (45):

$$c_z^{t+1} = \left(\frac{\left(c_z^t \frac{q_{z+1}^t - q_z^t}{dz} + 2q^t \frac{c_{z+1}^t - c_z^t}{dz}\right)}{\theta_z^t} + \frac{2q^t Err_4}{\theta_z^t} + \frac{c_z^t Err_2}{\theta_z^t} - Err_5 - \frac{c_z^t Err_1}{\theta_z^t}\right)dt + c_z^t \tag{46}$$

where $Err_4$ and $Err_5$ are errors from the difference of $\frac{\partial c}{\partial z}$ and $\frac{\partial c}{\partial t}$, respectively.

When the coupled method is employed, Eq. (41) and Eq. (42) are solved simultaneously. Then, $c_z^{t+1}$ can be solved by integrating Eq. (41) and Eq. (42):

$$c_z^{t+1} = \left(\frac{\left(c^t \frac{q_{z+1}^t - q_z^t}{dz} + 2q^t \frac{c_{z+1}^t - c_z^t}{dz}\right)}{\theta_z^t} + \frac{2q^t Err_4}{\theta_z^t} + \frac{c_z^t Err_2}{\theta_z^t} - Err_5\right)dt + c_z^t \tag{47}$$

Analytically, $c_z^{t+1}$ from the segregated method (Eq. 46) has more errors ($-c_z^t \frac{Err_1}{\theta_z^t}$) than $c_z^{t+1}$ from the coupled method (Eq. 47) in this example. The extra error term illustrates that numerical errors from the soil water flow equation will be transferred to the isotope transport equation. Furthermore, the errors will be accumulated in each temporal step and could result in the oscillation in isotopic solutions. This may be one of the reasons that SiSPAT (Braud et al., 2005) and revised HYDRUS-1D

(Zhou et al., 2021) require the thickness of the first soil layer to be small enough to minimize the mass balance errors (analogy to $Err_1$) of soil water to the order of $10^{-16}$ (Zhou et al., 2021). Note that the example shown here is simplified. In reality, the transport of isotope species within subsurface soil layers are influenced by many processes including evaporation, infiltration, fractionation, diffusion, and dispersion. Errors from the isotope transport equations will be enlarged when all these processes

are integrated into the highly nonlinear partial differential equations. Therefore, it is important for a segregated method to control errors at the beginning of the simulation by choosing appropriate temporal and spatial step combinations (Zha et al., 2019). However, for the coupled method, the mass balance errors will not be transferred and accumulated between equations, which may be the reason that the MOIST model can accommodate greater spatial and temporal steps than existing models





under the theoretical tests.

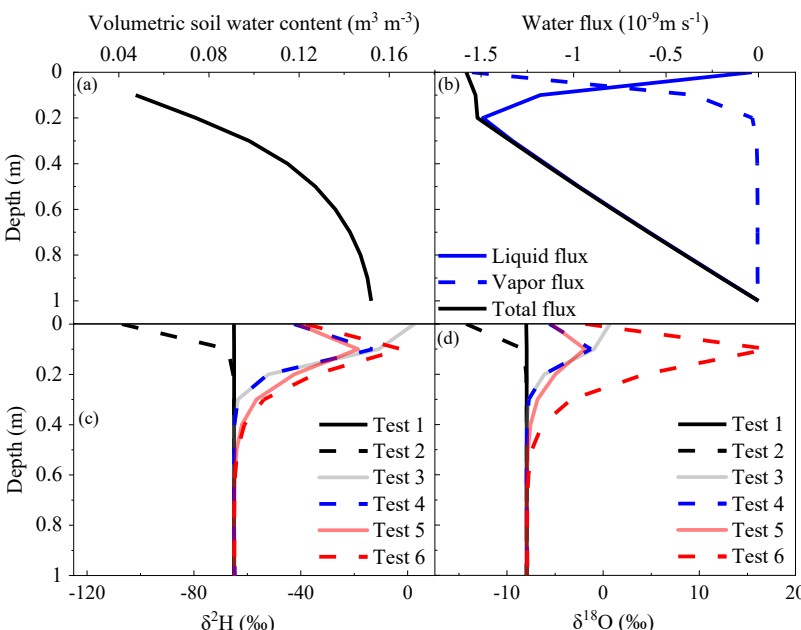


**Figure 12. Results from the theoretical tests performed at the end of the MOIST simulation, using a spatial step of 0.1 m. (a) Volumetric soil water content; (b) liquid and vapor flux; (c) soil $\delta^2H$ and (d) soil $\delta^{18}O$.**

## 4.2 Model Accuracy of the short-term simulation

The EPFL site experienced significant fractionation processes during isotope transport (Nehemy et al., 2021), providing an

excellent site for validating model performance on soil water and deuterium transport under realistic conditions. The *NSE*, $R^2$,

and *MAE* results showed that MOIST had a good performance in simulating soil water content (Table 4). For isotope transport,

the overall *MAE* was 11.84‰, which is acceptable in comparison to similar studies. Comparable *MAE* (10‰-20‰) was found

by Maloszewski et al. (2006), where the transport of deuterium was simulated in seven lysimeters with different soil materials.

Melayah et al. (1996b) also modeled the transport of deuterium in unsaturated soils under natural conditions and found the

*MAE* was approximately 20‰. Furthermore, the one-pore domain model employed by Sprenger et al. (2018) had an *MAE* of

15‰ for simulated deuterium. This error reduced to 5‰ when a two-region model was used, but this approach was not

considered in MOIST, which is a subject of future consideration.

## 4.3 Why MOIST has better performance under the long-term simulation?

Stumpp et al., (2012) presented an experiment, which was ideal to evaluate model performance over a long period. Generally,

the longer the simulation, the greater the potential of a model to fail. This is because a longer simulation period is more likely

to experience extreme upper boundary conditions, which could result in the oscillation of numerical solutions. More



importantly, numerical errors tend to accumulate throughout the simulation period (Finzel et al., 2016) and, thus, long-term simulations are better in verifying the accuracy and stability of a numerical model.

The $\delta^{18}O$ of seepage water remained stable within the first 150 days of the simulation (Fig. 11), illustrating that water transit time through the whole column was approximately 150 days. This is consistent with the simulated results from rHS (Stumpp et al. 2012) and rHZ (Zhou et al. 2021) models. However, MOIST and rHS underestimated $\delta^{18}O$ between the 1200[th] and 1500[th] day (Fig. 11). This underestimation of the simulated results occurred because between the 1050[th] and 1300[th] day, the rainfall isotopic signals were relatively enriched, and the total precipitation amount was large. Inevitably, a portion of the draining

precipitation carried these enriched signals to the bottom of the soil column through preferential flow, resulting in the enrichment of $\delta^{18}O$ in outflow. Therefore, because MOIST and rHS are piston flow based and have no consideration of preferential flow, the predicted $\delta^{18}O$ of drainage water was underestimated. For the period between the 1500[th] and 1735[th] day, MOIST fit the measurements better than the rHS model. The simulated minimum value from rHS for this period was more depleted than those of MOIST and of the measured values (Fig. 11). This contrast was observed because unlike the rHS model,

MOIST considered equilibrium fractionation, which resulted in the enrichment of $\delta^{18}O$ in soil water, as described in Test 3 (Fig. 5d).

    The rHZ model also considered fractionation but did not perform as well as MOIST (Table 6) for two reasons. Firstly, all the governing equations for soil water, heat, and isotope transport are coupled in MOIST, while the isotope transport equations of

rHZ are segregated. A fully coupled method is generally more accurate and robust than a segregated approach when solving incompressible fluid problems (Ammara and Masson, 2004; Pascau et al., 1996). This is because less numerical errors will be accumulated from the coupled method, as mentioned in Sec. 4.1. Although segregated methods ignore interactions among variables, it still has the advantage of faster convergence and smaller memory usage than coupled methods (COMSOL, 2022).

Secondly, the rHZ model is vertex-centered and, thus, calculates the isotopic flux at the upper boundary using the isotopic compositions of the first node (at the soil surface) directly. However, MOIST is based on a cell-centered scheme, where the calculated isotopic flux at the upper boundary not only depends on the isotopic gradient at the air-soil interface, but also includes the isotopic transport of liquid and vapor phases between the soil surface and subsurface layer (Eq. 28). We incorporated this feature into MOIST because isotopic transport at the soil-air interface can be influenced by both atmospheric

isotopic composition (Zhou et al., 2021) and isotopic signals from soil water (Haverd and Cuntz, 2010). Using a cell-centered scheme, it is possible to obtain a more accurate isotope mass balance at the soil-air interface when diffusive and dispersive isotopic fluxes are considered. For the vertex-centered scheme; however, isotopic transport at the soil-air interface may be dominated by atmospheric isotopic signals and be less affected by isotopic compositions of subsurface soil water. Although

both vertex- and cell-centered schemes are widely used (Crevoisier et al., 2009; Huber and Helmig, 2000; Manzini and Ferraris, 2004; Ross, 2003; Zha et al., 2016), their performances under different soil textures and boundary conditions may vary (Farthing and Ogden, 2017). The soil used in the long-term study was coarse-textured and, therefore, according to previous studies the cell-centered scheme may be more accurate (Fallah, 2004; Kollet and Maxwell, 2006; Ferguson and Turner, 1995; A. Szymkiewicz and Helmig, 2011; Adam Szymkiewicz et al., 2015). This is because the vertex-centered scheme may underestimate the inter-nodal conductivity of coarse textured soils during infiltration processes (Delis et al., 2011;

Szymkiewicz and Helmig, 2011).

**5. Conclusion**

We developed MOIST, a novel soil water and isotope transport model using MATLAB programming language. MOIST is unique in that it solves water, vapor, heat, and isotope transport simultaneously. The new model successfully passed well-known theoretical tests, and semi-analytical solutions. Even when using large spatial steps (0.1 m), MOIST showed good

stability and numerical efficiency. We also tested the model against well-controlled short- and long-term lysimeter studies. The model showed good agreement between measured and predicted values for the short-term simulations at EPFL in Switzerland and outperformed the rHS and rHZ models in the long-term simulations at HBLFA Raumberg-Gumpenstein in Austria. It can be concluded that MOIST is a powerful tool for simulating one-dimensional isotope transport within soil. Moreover, the model can be customized according to different requirements as compared to other models. As such, the adopted equations and

parameters can be easily updated as our knowledge about isotope fractionation and transport continues to expand, making MOIST a suitable tool for both current and future exploration.

**Code and data availability**

The source code of MOIST can be found at https://github.com/HAN-2/MOIST. The short-term and long-term measurements can be found at https://zenodo.org/record/4037240#.Y029l3bMKUk and https://www.pc-progress.com/en/Default.aspx?h1d-

lib-isotope, respectively.

**Author contribution**

HF and BS designed the research, developed the model code, and performed the simulations. HF prepared the manuscript with contributions from all co-authors.

**Competing interests**

The authors declare that they have no conflict of interest.





**Acknowledgments**

We thank Andrew Ireson for the instruction of numerical modeling and Yuki Sunakawa for the discussions.

**Financial support**

This research was supported by the Natural Sciences and Engineering Research Council of Canada and the China Scholarship

Council (grant no. 201906300103)

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
