# Peer review of "MOIST: a MATLAB-based fully coupled one-dimensional isotope and soil water transport model"

_Hydrology and Earth System Sciences, 2022_

## Author Comment (AC1)

Dear reviewer,

Thank you for your detailed comments. The following are our detailed responses to each of your comments. In the following, the reviewer's comments are highlighted in boldface and our responses are in normal text.

('Equation' refers to the equation in manuscript, while 'Eq' refers to the equations in this document.)

***'Equation (1) is the water content-based Richards equation. It is well known that this kind of formulation cannot handle saturated problems and is not well posed at the interface between two layers, because water content is discontinuous. The mixed form of Richards equation should be used.'***

We think Equation (1) in the manuscript is the mixed form (with combination of Equation (3)). We wrote this as the general governing equation. However, our MATLAB code solves the head-based Richards equation, not the water-content based equation.

The head-based form and the mixed form of Richards equation are both commonly used for modeling water flow in porous media. We choose the head-based form because the head is continuous across the soil interface (especially for 3 or more layered soils) (Zha et al., 2019). In addition, many popular models, such as SWAP (van Dam and Feddes, 2000), are also based on head-based Richards' equation:

$$c(\varphi)\frac{\partial \varphi}{\partial t} - \frac{\partial}{\partial z}\left(K(\varphi)\left(\frac{\partial \varphi}{\partial z} - 1\right)\right) = 0 \tag{1}$$

where $c(\varphi)$ equals $\frac{\partial \theta}{\partial \varphi}$, and known as moisture capacity.

The mixed form of Richards' equation is no doubt a better choice. Because $c(\varphi)\frac{\partial \varphi}{\partial t}$ may not numerically equal $\frac{\partial \theta}{\partial t}$ (Celia et al., 1990; Clark et al., 2021), $c$ (from Eq. (1)), could introduce mass balance errors. However, the mass balance of head-based form can be significantly improved by a second-order approximation to the time derivative (Celia et al., 1990) and effectively controlled by adaptive time-stepping schemes (Ireson et al., 2023). Given our solvers adopt adaptive time stepping schemes, our program meets the mass balance requirement.

To reduce confusion, we will change Equation (1) in the manuscript to a head-based equation in the revised manuscript, which will not affect the remaining part of the manuscript. We will also add the above to our discussion section.

***'The heat transfer equation (2) is not correct. Csoil depends on the water content and should be embed in the time derivative.'***

We think our equation (2) is correct. The proof is provided below. As you indicated that $C_{soil}$ is a function of soil water content, and the heat transport equation can be written as:

$$\frac{\partial C_{soil}(\theta)T}{\partial t} = \frac{\partial}{\partial z}\left(k_H \frac{\partial T}{\partial z}\right) - C_w \frac{\partial qT}{\partial z} - C_w ST \tag{2}$$

However, this equation can be further rewritten as:

$$C_{soil}(\theta)\frac{\partial T}{\partial t} + TC_w \frac{\partial \theta}{\partial t} = \frac{\partial}{\partial z}\left(k_H \frac{\partial T}{\partial z}\right) - C_w q \frac{\partial T}{\partial z} - C_w T \frac{\partial q}{\partial z} - C_w ST \tag{3}$$

by combining:

$$\frac{\partial \theta}{\partial t} = -\frac{\partial q}{\partial z} - S \tag{4}$$

Eq. (3) can be simplified:

$$C_{soil}\frac{\partial T}{\partial t} = \frac{\partial}{\partial z}\left(\frac{k_H \partial T}{\partial z}\right) - C_w q \frac{\partial T}{\partial x} \tag{5}$$

Therefore, the heat transport equation in our manuscript is correct. It is just another form of the same expression.

*'More information should be provided concerning the solvers (ode113, ode23tb).'*

Thank you for your comments. Details about ode113 and ode23tb in the manuscript can be found on the MATLAB official website (https://www.mathworks.com/help/matlab/math/choose-an-ode-solver.html). To address your comment, we will add a brief description about the two solvers in the revised manuscript, put citation and the internet link to help readers understand the solvers.

*'The tests 1 to 6 are very qualitatively discussed. Only different types of processes are checked. Physical processes can be verified but it does not mean that the computed variables and the process kinetics are correct. Moreover, these tests are development tests. They do not provide any new information on processes and therefore, should not be part of the manuscript. It is expected that models overcome these kinds of tests before publication.'*

Thank you for your comments. These theoretical test results are included to keep consistency with previous isotope modelling studies. As mentioned in the manuscript, these tests are necessary for an isotope model because they can validate the accuracy and stability of the numerical model. Besides, these tests are included in the Sispat (Braud et al., 2005) and recent updated HYDRUS (Zhou et al., 2021). However, we also agree to your opinion, these tests do not provide any new information and the model should pass these tests before publication. Therefore, we will put them to appendix as necessary.

*"L479-480: the reason for poor MAE value is unclear to me."*

Figure 10a showed that the measured peak value of $\delta^2 H$ does not match the simulation, which result in a large *MAE* at the top 0.1 m. Except this peak point, MOIST had good estimations on the remaining points.

The large *MAE* could be related to the heterogeneity of the soil column/flow paths. For example,

the sampling area on the 11$^{th}$ day participated less in the flow processes. Besides, evaporation results in the fractionation of deuterium at the top 0.1 m layer. Therefore, the simulated value does not match the measurement and lead to a large *MAE*.

*"L521-540: The analysis of the difference between fully coupled or sequential approach (segregation) is convincing but it applies for an explicit time scheme discretization whereas HYDRUS and SiSPat use an implicit scheme. Moreover, the flow equation is written in terms of water content for MOIST, the other codes are using pressure based or a mixed form of Richards equation."*

Thank you for your comments. The segregated method, either the implicit scheme or the explicit scheme, may introduce more errors than fully coupled method. The implicit scheme may have better performance than explicit one because the former is more stable. Both implicit and explicit schemes, solve PDEs numerically, therefore, there will be always errors accumulation in either of the two schemes, and therefore still can accumulate errors if a set of partial differential equations are solved sequentially. The coupled method, however, can reduce the error accumulation by solving a set of PDEs simultaneously. In out manuscript, an explicit example was used for easier understanding of the error difference between segregated and couple method.

Again, we used a head-based Richard equation, as can be seen from the MOIST source codes.

*'L608-610: The discussion about boundary conditions and intermodal conductivity is very popular. There are key papers not cited in the manuscript that review some of the techniques (see for example Belfort et al.,. On equivalent hydraulic conductivity for oscillation–free solutions of Richards equation. Journal of Hydrology, 2013, 505, pp.202-217).'*

Thank you for your suggestion. We will cite these key papers in our revised manuscript.

*"MOIST was used to simulate two types of experiments and the authors concluded that MOIST is more accurate and reliable. This is not supported by the provided results. These results only show that MOIST might be better calibrated not that the numerical scheme – fully coupled- is better than other schemes. Parameters used by MOIST and the other models should be given."*

We did not calibrate our model for both datasets. We used the parameters for the long-term experiment site that is provided by Stumpp et al. (2012) and also used by Zhou et al., (2021). Therefore, the model parameters are identical between our study and Stumpp et al. (2012) and Zhou et al. (2021). These parameters can be found in Table 4 from Stumpp et al. (2012) and the comparisons are showed in Figure 11 and Table 6 in the manuscript. We believe these comparisons support that MOIST has better performance mainly because the numerical scheme is different (segregated vs. coupled). However, in the revised manuscript, we will state the limitation that we only used two datasets for testing our model. More extensive verification is needed.

*"The comparisons do not provide any information on the code accuracy and efficiency. To demonstrate the 'excellent performance of the MOIST,' the authors should compare their code with other existing codes (for example looking at breakthrough curves at*

*different locations) and check detailed mass balances, time and space discretization sensitivity and computer time."*

Our tests are not exhaustive. But we do have most elements as you suggested for testing accuracy and efficiency. As mentioned above, we compared our code to different version of Hydrus under the long-term simulation (data come from Stumpp et al. (2012)). The simulated isotopic composition of outflow from a lysimeter is equivalent to the breakthrough curve simulation. In addition, the short-term experiment in EPFL has a spike treatment (irrigate water with a high concentration in a short time), then the simulated isotopic concentration at different depths can also be treated as breakthrough curves simulation, and the accuracy is checked by measured isotopic compositions at different soil depth.

Moreover, the first part of discussion (4.1) illustrated the sensitivity of MOIST on space discretization. As compared to Sispat, Soil-litter-ISO, and revised HYDRUS, MOIST can pass these theoretical tests by using a ten-times larger spatial step. The test 1 (Figure 12c and 12d in the manuscript) also showed that MOIST has a good mass balance performance and, the final isotope distribution being a straight line (test 1), especially at the top of the soil column (Braud et al., 2005).

We will state in the revised manuscript that our tests are not exhaustive, and more tests are needed.

*Reference*

Braud, I., Bariac, T., Gaudet, J. P., and Vauclin, M.: SiSPAT-Isotope, a coupled heat, water and stable isotope (HDO and H $218$O) transport model for bare soil. Part I. Model description and first verifications, J Hydrol (Amst), 309, 277–300, https://doi.org/10.1016/j.jhydrol.2004.12.013, 2005.

Celia, M. A., Bouloutas, E. T., and Zarba, R. L.: A general mass-conservative numerical solution for the unsaturated flow equation, Water Resour Res, 26, 1483–1496, https://doi.org/10.1029/WR026i007p01483, 1990.

Clark, M. P., Zolfaghari, R., Green, K. R., Trim, S., Knoben, W. J. M., Bennett, A., Nijssen, B., Ireson, A., and Spiteri, R. J.: The numerical implementation of land models: Problem formulation and laugh tests, J Hydrometeorol, 22, 1627–1648, https://doi.org/10.1175/JHM-D-20-0175.1, 2021.

van Dam, J. C. and Feddes, R. A.: Numerical simulation of infiltration, evaporation and shallow groundwater levels with the Richards equation, J Hydrol (Amst), 233, 72–85, https://doi.org/https://doi.org/10.1016/S0022-1694(00)00227-4, 2000.

Ireson, A. M., Spiteri, R. J., Clark, M. P., and Mathias, S. A.: A simple, efficient, mass-conservative approach to solving Richards' equation (openRE, v1.0), Geosci Model Dev, 16, 659–677, https://doi.org/10.5194/gmd-16-659-2023, 2023.

Zha, Y., Yang, J., Zeng, J., Tso, C. M., Zeng, W., and Shi, L.: Review of numerical solution of Richardson–Richards equation for variably saturated flow in soils, Wiley Interdisciplinary Reviews: Water, 6, 1–23, https://doi.org/10.1002/wat2.1364, 2019.

---

## Author Comment (AC2)

Dear reviewer,

Thank you for your comments, the following are our responses. Comments are highlighted in boldface and our responses are in normal text.

('Equation' refers to the equations in manuscript, while 'Eq' refers to the equations in this document.)

***The paper has some interesting aspects. A fully coupled isotope transport model in the soil-plant-atmosphere continuum needs to be improved in the recent literature. Some approximations have been made to simulate isotope transport in soil using HYDRUS for example, but the results could be better. This is an intricate problem that must consider water content and movement influences water temperature, and both influence isotope transport and fractionation, and temperature may also affect water movement.***

***The paper claims to solve the transport equations simultaneously. I would like to see the numeric scheme that shows this back-forward process. And yet more information about it is needed. Unfortunately, numerical implementation has only one paragraph.***

Thank you for your comments. Indeed, we did not introduce how to solve soil water, heat, and isotope transport equations simultaneously, and the solvers we used in detail. We will provide this information as appendix in the revised version of manuscript.

The transport equations of soil water, heat, and isotopes are:

$$\frac{\partial \theta}{\partial t} + \frac{\partial \theta_v}{\partial t} = -\frac{\partial q}{\partial z} - S \tag{1}$$

$$C_{soil}\frac{\partial T}{\partial t} + \rho\lambda_E\frac{\partial \theta_v}{\partial t} = -\frac{\partial q_T}{\partial z} \tag{2}$$

$$\frac{\partial\left(C_{il}(\theta+\alpha\theta_v)\right)}{\partial t} = -\frac{\partial q_i}{\partial z} - C_{il}S \tag{3}$$

where $\theta$ and $\theta_v$ are the soil water content and equivalent liquid water content (m³ m⁻³), respectively; $q$ is the water flux (m s⁻¹); $S$ is the sink term (s⁻¹); $C_{soil}$ is the soil heat capacity (J m⁻³ K); $T$ is the temperature; $\lambda_E$ is the latent heat of vaporization (J kg⁻¹); $\rho$ is the water density (kg m⁻³); $q_T$ is the heat flux (J m⁻² s⁻¹); $C_{il}$ is the isotopic concentration of soil water (kg m⁻³); $\alpha$ is the equilibrium fractionation coefficient ($\alpha^*$ in the manuscript); $q_i$ is the isotopic flux (kg m⁻² s⁻¹). Detailed description of Eq. (2) can be referred to Appendix A.

The equivalent liquid water content $\theta_v$ $\left(m^3_{liquid\,water}\,m^{-3}_{soil}\right)$ can be expressed by pore space within soil $\theta_s$-$\theta$ $\left(m^3_{air}\,m^{-3}_{soil}\right)$ and the saturated vapor concentration within soil air is expressed as $Cv_{sat}$ $\left(m^3_{liquid\,water}\,m^{-3}_{air}\right)$:

$$Cv_{sat} = \frac{m^3_{liquid\,water}}{m^3_{air}} = \frac{\frac{mass_{liquid\,water}}{\rho_{liquid\,water}}}{\frac{mass_{air}}{\rho_{air}}} = \frac{\frac{mass_{vapor}}{\rho_{liquid\,water}}}{\frac{mass_{air}}{\rho_{air}}} \tag{4}$$

where $\theta_s$ is the saturated soil water content (m³ m⁻³); $m^3_{liquid\,water}$ and $m^3_{air}$ are the volume of liquid water (m³) and air (m³) within soil pore space; $mass_{liquid\,water}$ and $mass_{air}$ are the mass of liquid water (kg) and mass of air (kg) in the soil pore space; $\rho_{liquid\,water}$ and $\rho_{air}$ are the density of liquid water (kg m⁻³) and air (kg m⁻³), respectively; $mass_{vapor}$ is the mass of vapor (kg) within soil

pore space.

Then, the ideal gas law can be incorporated into Eq. (4):

$$Cv_{sat} = \frac{\frac{mass_{vapor}}{\rho_{liquid\ water}}}{\frac{mass_{air}}{\rho_{air}}} = \frac{\rho_{air}}{\rho_{liquid\ water}} \frac{P_{vapor_{sat}} M_{water}}{P_{air} M_{air}} \tag{5}$$

where $P_{vapor_{sat}}$ and $P_{air}$ are the saturated vapor pressure (kpa) and air pressure (kpa), respectively; $M_{water}$ and $M_{air}$ are the mole weight of water (kg mol$^{-1}$) and air (kg mol$^{-1}$), respectively.

Eq. (5) can be further simplified by applying the ideal gas law again on $P_{air}$:

$$Cv_{sat} = \frac{\rho_{air}}{\rho_{liquid\ water}} \frac{P_{vapor_{sat}} M_{water}}{P_{air} M_{air}} = \frac{\rho_{air}}{\rho_{liquid\ water}} \frac{P_{vapor_{sat}} M_{water}}{\rho_{air}\frac{R}{M_{air}}T M_{air}} \tag{6}$$

where $R$ is the ideal gas constant (J mol$^{-1}$ K$^{-1}$). Similarly, the unsaturated vapor concentration in soil pore space in terms of equivalent liquid water content, $Cv$, is given by:

$$Cv = \frac{\rho_{air}}{\rho_{liquid\ water}} \frac{P_{vapor_{sat}} M_{water}}{P_{air} M_{air}} = \frac{\rho_{air}}{\rho_{liquid\ water}} \frac{P_{vapor} M_{water}}{\rho_{air}\frac{R}{M_{air}}T M_{air}} \tag{7}$$

Saturated vapor pressure, $P_{vapor_{sat}}$, can be calculated by Tetens formula (Ham, 2015). Then, $Cv_{sat}$ is written as:

$$Cv_{sat} = \frac{0.61078 e^{\frac{17.269T}{T+237.29}} M_{water}}{\rho_{liquid\ water} RT} \tag{8}$$

Considering the influence of variation of both soil water content and temperature on the relative humidity, $h_r$ is given by (Philip, 1957):

$$h_r = e^{\frac{M_w hg}{RT}} \tag{9}$$

Besides, according to the definition of $h_r$:

$$h_r = \frac{P_{vapor}}{P_{vapor_{sat}}} \tag{10}$$

which can be rewritten by combining Eqs. (6) and (7):

$$h_r = \frac{Cv}{Cv_{sat}} \tag{11}$$

Therefore, the equivalent water content of the volumetric water vapor content, $\theta_v$, can be written as:

$$\theta_v = (\theta_s - \theta) Cv_{sat} h_r \tag{12}$$

Introducing Eq. (12) to Eqs. (1) - (3):

$$\frac{\partial \theta}{\partial t} + \frac{\partial((\theta_s - \theta)\ Cv_{sat} h_r)}{\partial t} = -\frac{\partial q}{\partial z} - S \tag{13}$$

$$\frac{C_{soil}\partial T}{\partial t} + \frac{\rho \lambda_E \partial((\theta_s - \theta) Cv_{sat} h_r)}{\partial t} = -\frac{\partial q_T}{\partial z} \tag{14}$$

$$\frac{\partial\left(C_{il}\left(\theta + \alpha Cv_{sat} h_r(\theta_s - \theta)\right)\right)}{\partial t} = -\frac{\partial q_i}{\partial z} - C_{il}S \tag{15}$$

Note that the head-based Richards' equation is used in our model, $\frac{\partial h}{\partial t}$, $\frac{\partial T}{\partial t}$, and $\frac{\partial C_{il}}{\partial t}$ are isolated to solve Eq. (13), (14), and (15) for $h$, $T$, and $C_{il}$ at each time step simultaneously. Since $Cv_{sat}$ is the function of $T$ (Eq. (8)), $h_r$ is the function of $h$ and $T$ (Eq. (9)), and $\alpha$ is the function of $T$ (Equation. (18)), the analytical expressions of $\frac{\partial h}{\partial t}$, $\frac{\partial T}{\partial t}$, and $\frac{\partial C_{il}}{\partial t}$ can be written as:

$$\frac{\partial h}{\partial t}=\frac{1}{A}\left(-\frac{\partial q_T}{\partial z}-B\frac{\left(-\frac{A}{C}\left(\frac{\partial q}{\partial z}+S\right)+\frac{\partial q_T}{\partial z}\right)}{\frac{D}{C}A-B}\right) \tag{16}$$

$$\frac{\partial T}{\partial t}=\frac{-\frac{A}{C}\left(\frac{\partial q}{\partial z}+S\right)+\frac{\partial q_T}{\partial z}}{\frac{D}{C}A-B} \tag{17}$$

$$\frac{\partial C_{il}}{\partial t}=\frac{F}{E} \tag{18}$$

with coefficients A to E:

$$A=\rho\lambda_E\left((\theta_s-\theta)\ Cv_{sat}\frac{\partial h_r}{\partial h}-Cv_{sat}h_r\frac{\partial\theta}{\partial h}\right) \tag{19}$$

$$B=C_{soil}+\rho\lambda_E\left((\theta_s-\theta)Cv_{sat}\frac{\partial h_r}{\partial T}+(\theta_s-\theta)h_r\frac{\partial Cv_{sat}}{\partial T}\right) \tag{20}$$

$$C=(1-Cv_{sat}h_r)\frac{\partial\theta}{\partial h}+(\theta_s-\theta)Cv_{sat}\frac{\partial h_r}{\partial h} \tag{21}$$

$$D=(\theta_s-\theta)Cv_{sat}\frac{\partial h_r}{\partial T}+(\theta_s-\theta)\ h_r\frac{\partial Cv_{sat}}{\partial T} \tag{22}$$

$$E=\theta+\alpha Cv_{sat}\ h_r\ (\theta_s-\theta) \tag{23}$$

$$F=-\frac{\partial q_i}{\partial z}-C_{il}S-C_{il}\frac{\partial\theta}{\partial h}\frac{\partial h}{\partial t}-$$

$$C_{il}\theta_s\left(Cv_{sat}h_r\frac{\partial\alpha}{\partial T}\frac{\partial T}{\partial t}+\alpha h_r\frac{\partial Cv_{sat}}{\partial T}\frac{\partial T}{\partial t}+\alpha Cv_{sat}\left(\frac{\partial h_r}{\partial h}\frac{\partial h}{\partial t}+\frac{\partial h_r}{\partial T}\frac{\partial T}{\partial t}\right)\right)+C_{il}\left(Cv_{sat}h_r\theta\frac{\partial\alpha}{\partial T}\frac{\partial T}{\partial t}+\alpha h_r\theta\right.$$

$$\left.\frac{\partial Cv_{sat}}{\partial T}\frac{\partial T}{\partial t}+\alpha Cv_{sat}\theta\left(\frac{\partial h_r}{\partial h}\frac{\partial h}{\partial t}+\frac{\partial h_r}{\partial T}\frac{\partial T}{\partial t}\right)+\alpha Cv_{sat}h_r\frac{\partial\theta}{\partial h}\frac{\partial h}{\partial t}\right) \tag{24}$$

Eqs. (13), (14), and (15) were transformed into a system of coupled ordinary differential equations by Eqs. (16)-(24). This system is solved by MATLAB solvers (ode113/ode23tb) simultaneously. The derivative vector ($\frac{\partial h}{\partial t}$, $\frac{\partial T}{\partial t}$, and $\frac{\partial C_{il}}{\partial t}$), having a length of the number of spatial discretization multiplied by three.

To construct the derivative vector, values from Eq. (17) were calculated firstly because they were also used in Eq. (16). Eq. (17) shows that temporal variation of temperature was influenced by $q$, $q_T$, $\theta$, and other parameters from coefficients $A$-$D$. Reversely, $T$ influences $h_r$ and $Cv_{sat}$ (Eqs. (19)-(22)) and further affect the water transport within soil (Eq. (16)).

Eq. (16) showed that temporal variation of $h$ was closely related to $T$ because $\frac{\left(-\frac{A}{C}\left(\frac{\partial q}{\partial z}+S\right)+\frac{\partial q_T}{\partial z}\right)}{\frac{D}{C}A-B}$ from Eq. (16) is $\frac{\partial T}{\partial t}$ (Eq. (17)). Besides, soil heat properties, such as soil heat capacity and latent heat of

vaporization (included in coefficients $A$ and $B$), were also involved in soil water (vapor) movement.

Eq. (24) shows that isotope transport was influenced by all the parameters coupled. Specifically, water transport (Eq. (16)) affects isotopic fluxes since isotopes were treated as solutes, while heat transport (Eq. (17)) had an influence on equilibrium fractionation coefficients and further on the isotopic concentration in soil water. Both water and heat transport affected $h_r$ and $Cv_{sat}$ in soil. Therefore, values for derivative vector construction from Eq. (18) were calculated based on Eqs. (16), (17), (23), and (24).

Finally, the derivative vector, along with the initial conditions and the time span were passed to the solvers. The solver then computed the solution of this system over the specified time span numerically. Numerical schemes of solvers are described below.

*Numerical scheme of ode113*

The ode113 solver uses an adaptive, variable-order, variable-step-size (VOVS) method. This is implemented with a variable order Adams-Bashforth-Moulton (ABM) method (ode113, 2023), which is a combination of an explicit types of the Adams-Bashforth (AB) and an implicit type of Adams-Moulton (AM) methods. Specifically, the AB method is used to estimate the solution at the new time step by taking multiple previous time steps into account, while the AM method is used to make corrections.

The ode113 can select automatically between the 1$^{st}$ and 13$^{th}$ order approximation (the highest order used appears to be 12) during the computation based on the estimation errors. This is helpful for minimizing the estimated errors and for achieving high efficiency in time. Moreover, the time step size is adjusted according to the estimation error. In this way, ode113 can handle a wide range of ODE problems with high accuracy and efficiency.

Therefore, ode113 can do a good job when the transport media is relatively uniform. However, ode113 is susceptible to numerical oscillation when hydraulic conductivities between layers differed greatly because of the adopted explicit scheme.

*Numerical schemes of ode23tb*

Ode23tb is a solver specifically designed for solving ODEs with highly oscillatory solutions (ode23tb, 2023), such as those arising from heterogeneity in hydraulic conductivities between soil layers. The 'tb' stands for that the solver combines a trapezoidal rule (sometimes referred as the second-order AM method (Adams methods, 2023)) with a 2$^{nd}$ order backward differentiation formula (BDF). Because of this, ode23tb is an efficient and accurate solver for stiff ODE systems, making it less susceptible to numerical instability.

Like ode113, ode23tb can adjust the step size automatically based on the estimated error and the oscillatory behavior of the solution. However, ode23tb is an implicit solver, making it more computationally expensive than other solvers. But because it adopts the trapezoidal BDF method, it is more efficient and accurate than other types of implicit methods, such as the fully implicit Euler method or the backward Euler method (Time integration, 2023). Therefore, ode23tb may work

better than ode113 when the soil physical properties are greatly differed between layers.

***Another question is, how do equations 41 and 42 take isotope fractionation from temperature variation into account?***

Thank you for your comments. Equation (41) and (42) in the manuscript are 'thought' experiment. They were used to illustrate the error difference between segregated and coupled methods. The segregated method may introduce more errors than coupled method because more errors could be accumulated as compared to the coupled method. The coupled method, however, can reduce the error accumulation by solving a set of equations simultaneously. In our manuscript, Equation (41) and (42) were used as an example for understanding the error difference between segregated and coupled method.

*Appendix A:*

The heat transport equation within soil is written as:

$$C_{soil}\frac{\partial T}{\partial t}+\rho\lambda_E\frac{\partial\theta_v}{\partial t} = \frac{\partial}{\partial z}\left(k_H\frac{\partial T}{\partial z}\right)-C_wq_l\frac{\partial T}{\partial z}-C_{vh}\frac{\partial q_vT}{\partial z}-\rho\lambda_E\frac{\partial q_v}{\partial z} \tag{A1}$$

Eq. (A1) can be rewritten by the chains rule:

$$C_{soil}\frac{\partial T}{\partial t}+\rho\lambda_E\frac{\partial\theta_v}{\partial t} = \frac{\partial}{\partial z}\left(k_H\frac{\partial T}{\partial z}\right)-C_wq_l\frac{\partial T}{\partial z}-C_{vh}q_v\frac{\partial T}{\partial z}-C_{vh}T\frac{\partial q_v}{\partial z}-\rho\,\lambda_E\frac{\partial q_v}{\partial z} \tag{A2}$$

Then:

$$C_{soil}\frac{\partial T}{\partial t}+\rho\lambda_E\frac{\partial\theta_v}{\partial t} = \frac{\partial}{\partial z}\left(k_H\frac{\partial T}{\partial z}\right)-(C_wq_l+C_{vh}q_v)\frac{\partial T}{\partial z}-(C_{vh}T+\rho\lambda_E)\frac{\partial q_v}{\partial z} \tag{A3}$$

where $C_wq_l+C_{vh}q_v$ and $C_{vh}T+\rho\lambda_E$ can be treated as constants within each layer, and Eq. (A3) is written as:

$$C_{soil}\frac{\partial T}{\partial t}+\rho\lambda_E\frac{\partial\theta_v}{\partial t} = -\frac{\partial}{\partial z}\left(-k_H\frac{\partial T}{\partial z}+Constant_1T+Constant_2q_v\right) \tag{A4}$$

where $Constant_1=C_wq_l+C_{vh}q_v$ and $Constant_2=C_{vh}T+\rho\lambda_E$.

Assuming $q_T=-k_H\frac{\partial T}{\partial z}+Constant_1T+Constant_2q_v$, then:

$$C_{soil}\frac{\partial T}{\partial t}+\rho\lambda_E\frac{\partial\theta_v}{\partial t} = -\frac{\partial q_T}{\partial z} \tag{A5}$$

Eq. (A5) is the same as Eq. (2).

**Reference**

Time integration: https://space.mit.edu/RADIO/CST_online/mergedProjects/DES/theory/time_integration.htm, last access: 28 February 2023.

Ham, J. M.: Useful equations and tables in micrometeorology, Micrometeorology in Agricultural Systems, 533–560, https://doi.org/10.2134/agronmonogr47.c23, 2015.

ode23tb: https://www.mathworks.com/help/matlab/ref/ode23tb.html, last access: 28 February 2023.

ode113: https://www.mathworks.com/help/matlab/ref/ode113.html, last access: 28 February 2023.

Philip, J. R.: Evaporation, and moisture and heat fields in the soil, Journal of Meteorology, 14, 354–366, https://doi.org/https://doi.org/10.1175/1520-0469(1957)014<0354:EAMAHF>2.0.CO;2, 1959.

Adams methods: https://web.mit.edu/10.001/Web/Course_Notes/Differential_Equations_Notes/node6.html, last access: 28 February 2023.

---

## Author Comment (AC3)

Dear reviewer,

We were so lucky to receive such a detailed and high-quality review from you. This is one of the best review comments we received in the last two years in my academic career, and it will substantially improve the quality of manuscript.

As such, we are happy to provide our responses below to each of your comments. Comments are highlighted in boldface and our responses are in normal text.
('Eq' refers to the equations in manuscript, while 'Equation' refers to the equations in this document.)

We believe our objective is to exhibit the robustness and accuracy of MOIST, rather than have new insights into existing experiments. While our validations are not exhaustive, we have compared MOIST with two revised versions of HYDRUS-1D, which were published by Stumpp et al. (2012) and Zhou et al. (2021), respectively. Figure 11 in the manuscript showed that the results obtained from MOSIT were comparable to those obtained using two versions of revised HYDRUS-1D, and the statistical information presented in Table 6 indicated that MOIST slightly outperforms than them under certain conditions. However, we acknowledge that more validations are necessary in the future.

Regarding to the sensitivity analysis on the choice of kinetic fractionation factor formulations, we have included this in Appendix A of this document and plan to put it in the discussion section of the revised manuscript.

**0/ It would be useful to provide a list of notations with the units.**

Thank you for your comments. We will provide a list of notations with the units as appendix in the revised manuscript.

**1/L39: do you really address this problem with the MOIST model?**

In the study of Haverd and Cuntz (2010), the heat flux was calculated as the sum of sensible and latent heat (Eq. A.24 from Haverd and Cuntz (2010)), without considering the variation of heat capacity of the liquid and vapor phases. We implemented these variations in our heat transport equation. However, this consideration was not originally addressed in our work; rather, we just highlighted a limitation in the heat transport equation used by Haverd and Cuntz (2010).

**2/ L57: I do not consider that a Matlab program is something fully accessible as paying for an expensive Matlab licence is necessary to use the developments**

Thank you for your comments. MATLAB is one of widely used computer programming environment. We chose MATLAB because the good performance of the ode solvers, as described in Appendix B. Once well-tested, we will consider migrating MOIST to an open-source language, such as Python, so that the program is available both in MATLAB and Python.

**3/ Eq. (4) Dv is not defined**

Thank you for your comments. We will define Dv in the revised manuscript.

**4/ Eq. (5) looks strange**

Thank you for pointing out a typo in the manuscript (our program used the correct equation). The Eq. (5) should be:

$$C_{soil}\frac{\partial T}{\partial t}+\rho\lambda_E\frac{\partial\theta_v}{\partial t}=\frac{\partial}{\partial z}\left(k_H\frac{\partial T}{\partial z}\right)-C_wq_l\frac{\partial T}{\partial z}-C_{vh}\frac{\partial q_v T}{\partial z}-\rho\,\lambda_E\frac{\partial q_v}{\partial z} \tag{1}$$

**5/ Eq (10) in Braud et al. (2005) (their Equation (9) has an additional term). Why do you neglect it (as in Haverd and Cuntz, 2010)?**

Thank you for your comments. We did not neglect the additional term. As shown below, the Cv term in Equation (2) includes the additional term in Braud et a. (2005). Considered the vapor movement within soil is dominated by the Fick's law:

$$q_v=-D_v\,\frac{\partial c_v}{\partial z} \tag{2}$$

Similarly, the isotope movement in vapor phase within soil can be written as:

$$q_{iv}=-D_{iv}\frac{\partial c_{iv}}{\partial z} \tag{3}$$

However, Equation. (3) can be further written as:

$$q_{iv}=-D_{iv}\frac{\partial c_{iv}}{\partial z}=-D_{iv}\frac{\partial(c_v\,\alpha\,c_{il})}{\partial z}=-D_{iv}\alpha\,c_{il}\frac{\partial c_v}{\partial z}-D_{iv}c_v\left(\alpha\frac{\partial c_{il}}{\partial z}+c_{il}\frac{\partial\alpha}{\partial z}\right) \tag{4}$$

Then, by incorporating $\frac{D_{iv}}{D_v}=\alpha_{diff}$ and Equation. (2):

$$q_{iv}=-\alpha_{diff}\alpha\,c_{il}q_v-D_{iv}c_v\left(\alpha\frac{\partial c_{il}}{\partial z}+c_{il}\frac{\partial\alpha}{\partial z}\right) \tag{5}$$

Therefore, the isotope movement in vapor phase consists of a convection term ($-\alpha_{diff}\alpha\,c_{il}q_v$) and a diffusion term ($-D_{iv}c_v\,\alpha\frac{\partial c_{il}}{\partial z}$), as shown in the Braud et al. (2005).

**6/ Eq. (12) there are several options in the literature for the specification of nD. Why do you chose this formulation?**

Thank you for your comments. We chose nD equation from Melayah et al. (1996) because it describes the isotopic fractionation, which is caused by diffusion, in both wet and dry soil. In addition, the sensitivity analysis (Appendix A) showed that the MOIST is not sensitive to the choice of formulations when considering kinetic fractionation. Moreover, Braud et al. (2005) have stated that there are no objective criteria for selecting one formulation over others. Nevertheless, in the future, we can easily update this formulation in the MOIST source code with a more reasonable one, if needed.

*7/ Eq. (24) and (25) (even if taken from Haverd and Cuntz, 2010) are strange as they are already provided in a discretized form, contrarily to the other equations. Same comment for Eq. (27)*

Thank you for your comments. We presented the continuous form in a discretized form (Eq. (24), (25), and (27)) were written in a discretized form because they arethey are only used in the top half layer for upper boundary solutions. However, they should be in the continuous form. We will fix this problem we will add the continuous form in the revised manuscript.

*8/ section 2.1.4: the formulation of the boundary conditions for the various equations is very important for getting correct results. The authors put much effort in fully coupling the equations in the soil but put much less attention in the formulation of the boundary conditions. The choices made would require more justifications even if they seem to be similar to Haverd and Cuntz (2010).*
*More generally, it seems that the authors have recoded the Haverd and Cuntz (2010) model without considering the litter. It would be relevant to say it if it the truth.*

We utilized the equations of energy, water, and isotope mass conservation at the air-soil interface proposed by Haverd and Cuntz (2010) but did not consider the litter layer at the current stage of our work. MOIST is based on cell-centered numerical method, which requires us to consider the water content, temperature and isotopic compositions at soil-air interface and the fluxes within the top layer. This means that the fluxes from the center of the topsoil layer to the soil-air interface are balanced by the fluxes from soil-air interface to the atmosphere. The upper boundary calculation method from Haverd and Cuntz (2010) met our requirements, so we recoded the upper boundary from 'Soil-Litter-Isotope' model (Haverd and Cuntz, 2010). We will make this clearer in the revised manuscript.

*162. Should be Eq. (24) and (25)?*
Sorry for the typo. They should be Eq. (24) and (25). We will address this in the revised manuscript.

*9/ L191: should be Eq. (19)?*
Thanks. It should be Eq. (19). We will correct it in the revised manuscript.

*10/ the description of the numerical implementation is too short. What are the variables that you are computing? I would be curious to see the three discretized coupled equations in order to see where is the benefit of having them fully coupled. In other words, on which system of equations do you implement the Matlab solver (in the equations presently given in the paper, there are more than three unknown sets of variables).*

Thank you for your comments. We added detailed descriptions about the numerical implementation in Appendix B.

*11/ L194: when you use the soil water pressure as variable, it is continuous at the interface between layers with different hydraulic properties, which limits the problem you mention here.*

We agree. Our intention was to state that when physical properties are different between soil layers, the hydraulic conductivity can vary drastically at the layer interface, leading to potential oscillation problems. We will provide a more accurate statement in the revised manuscript.

**11/ L220. The reference with the DOI of the dataset should appear in the reference list, not only in the text. This is a reference like a standard paper and it should be cited as such.**

Thank you for your comments. We will cite the dataset correctly in the revised manuscript.

**12/ Eq. (34) and (35): did you checked the MOIST model behavior using the Van Genuchten (1980) model for the retention curve and the Brooks and Corey (1964) model for the hydraulic conductivity as done in Braud et al. (2005)?**

Thank you for your comments. We will add this information (like Table 2 in Braud et al., 2005 and Fig 3 in Zhou et al., 2021) in the revised manuscript.

**13/Figure 5, 6, 7: You could zoom on the 0-0.5 cm layer and avoid the 0 y-axis at the very top of the figure to get more legible figures. Furthermore, the Test case 3 is almost invisible (at least on a printed version).**

Thank you for your comments. We will update these figures in the revised manuscript.

**14/Section 3.2: graphical comparison is fine, but I would like to see a comparison between the simulated peak concentration and slope of the oxygen 18 – deuterium relationships (see Braud et al., 2005, Tables 3 and 5). This would provide a more robust evaluation of the model performance.**

Thank you for your comments. We will add this information in Section 3.2 in the revised manuscript.

**15/ the demonstration from Eq. (43) to (47) would require more details to be fully understandable.**

Thank you for your comments. Eq. (41) and (42) in the manuscript are 'thought' experiment. They were used to illustrate the error difference between segregated and coupled methods. The segregated method may introduce more errors than coupled method because more errors could be accumulated as compared to the coupled method. The coupled method, however, can reduce the error accumulation by solving a set of equations simultaneously. In our manuscript, Eq. (41) and (42) were used as an example for understanding the error difference between segregated and coupled method.

The example in the manuscript:

$$\frac{\partial \theta}{\partial t} = \frac{\partial q}{\partial z} \tag{6}$$

$$\frac{\partial (c\theta)}{\partial t} = 2\frac{\partial (qc)}{\partial z} \tag{7}$$

Equation (7) can be rewritten by chains rule:

$$c\frac{\partial \theta}{\partial t} + \theta\frac{\partial c}{\partial t} = 2\ \left(q\frac{\partial c}{\partial z} + c\frac{\partial q}{\partial z}\right) \tag{8}$$

If segregated method is used to solve these two equations for $\theta$ and $c$, they should be written discrete form in both spatial and temporal (explicit is the simplest to understand):

$$\frac{\theta_z^{t+1} - \theta_z^t}{\Delta t} = \frac{q_{z+1}^t - q_z^t}{\Delta z} \tag{9}$$

However, $\frac{\partial \theta}{\partial t}$ is not equal to $\frac{\theta_z^{t+1} - \theta_z^t}{\Delta t}$ within a small time step and this is true between $\frac{\partial q}{\partial z}$ and $\frac{q_{z+1}^t - q_z^t}{\Delta z}$ within a small spatial step. Therefore, error terms should be introduced to balance Equation (9), which are Err1 and Err2 in Eq. (43). Then $\theta$ can be solved in Equation (9) and further used to solve $c$ in Equation (8).

In the coupled method, Equation (8) can be rewritten by combining Equation (6):

$$\theta\frac{\partial (c)}{\partial t} = 2\ \left(q\frac{\partial (c)}{\partial z} + c\frac{\partial (q)}{\partial z}\right) - c\frac{\partial (q)}{\partial z} \tag{10}$$

Rewritten Equation (10) in discrete form can obtain Eq. (47) in the manuscript. Compared to segregated method, the coupled method can potentially reduce numerical errors by analytically inserting Equation (6) into Equation (10). Therefore, the coupled method should be more accurate than segregated method on solving a set of partial differential equations.

*16/ Figure 12 is not cited in the paper. Furthermore, if the purpose of the figure were to demonstrate that the coarser vertical spatial resolution provides results as accurate as in Figure 5, it would be more informative to show the difference between both simulations. Visually, it seems that the peak is simulated deeper in Figure 12 than in Figure 5, which would not be very satisfactory as the peak is located at the evaporation front.*

We forgot to cite Fig 12 at the end of line 554.

Regarding to the discrepancy in peak locations between Fig 12 and Fig 5, this is because when the coarse spatial step is used, some surface information could be lost. For example, in Fig 12, the soil water profile is also different from that in Fig 5 because we cannot capture the drying layer when the spatial step is larger than its thickness. This also explains why the peak locations in the isotope profiles are deeper in Figure 12, as the layer thickness is 0.1 m, which is larger than the location of the peak (around 0.05 m). Therefore, with larger spatial discretization, more surface information may be lost. However, MOIST still reflects the correct trend even under with coarse spatial discretization.

*17/ L588-593: this paragraph is not supported by the results. Other reasons than solving or not the fully coupled equations could explain discrepancies between the models, one of them being the specification of the boundary conditions or the specification of the kinetic fractionation factor, that may be different in the different models.*

Agreed. Indeed, there are many reasons could be related to the discrepancies between the model

because MOIST and HYDRUS have many differences. The upper boundary will play an important role, especially the water, heat, and isotopic fluxes vary constantly under field simulations. We will this in the revised manuscript.

***18/ L595-610: the question of the numerical method (cell-centered versus vertex-centered) is strange here as the method used in the MOIST model has not been presented before.***

Thank you for your comments. We will include the cell-centered method in method section.

**Appendix A**

According to Braud et al. (2005), we conducted the sensitive analysis on various formulations of the kinetic fractionation factor ($\alpha_k$) to calculate upper boundary conditions of isotope transport under non-saturated and non-isothermal conditions. These conditions are chosen because they are closer to reality than saturated and isothermal conditions.

The formulations used in five cases are described below.

Case 1:

$$\alpha_k = \frac{D_v}{D_{iv}} \tag{A1}$$

Case 2:

$$\alpha_k = \left(\frac{D_v}{D_{iv}}\right)^{nk} \tag{A2}$$

where $nk$ is calculated by:

$$nk = \frac{(\theta_{surface} - \theta_r)0.5 + (\theta_s - \theta_{surface})}{\theta_s - \theta_r} \tag{A3}$$

where $\theta_{surface}$ is the soil water conent at soil surface (m$^3$ m$^{-3}$); $\theta_r$ is residual soil water content (m$^3$ m$^{-3}$); $\theta_s$ is the saturated soil water content (m$^3$ m$^{-3}$).

Case 3:

$$\alpha_k = 1 \tag{A4}$$

Case 4:

$$\alpha_k = 1 + n_k \left(\frac{D_v}{D_{vi}} - 1\right)\frac{r_{am}}{r_a} \tag{A5}$$

where $r_a$ is the sum of $r_{am}$ and $r_{aT}$; $r_{am}$ and $r_{aT}$ are turbulent and molecular resistances to the water vapor transport. $r_{am}$ can be calculated by (Brutsaert, 1982):

$$if\ \frac{u^* Z_{om}}{v} \leq 1$$

$$r_{am} = 13.6 \left(\frac{v}{D_v}\right)^{\frac{2}{3}} \tag{A6}$$

$$if\ \frac{u^* Z_{om}}{v} \geq 1$$

$$r_{am} = 7.3 \left(\frac{u^* Z_{om}}{v}\right)^{\frac{1}{4}}\left(\frac{v}{D_v}\right)^{\frac{1}{2}} \tag{A7}$$

where $u^*$ is the friction velocity, which is 0 in this case because the wind speed is 0 m s$^{-1}$; $Z_{om}$ is the roughness length for momentum (m), $v$ is the air kinematic viscosity (m$^2$ s$^{-1}$).

Case 5:

$$\alpha_k = 1 \tag{A8}$$

$$n_D \neq 1 \tag{A9}$$

where $n_D$ is calculated by:

$$n_D = 0.67 + 0.33 exp(1 - \frac{\theta_{surface}}{\theta_r}) \qquad (A10)$$

Table A1 was formatted as Table 6 from Braud et al. (2005) and our results were found to be similar. However, calculated maximum $\delta^2H$ values were slightly smaller than those reported by Braud et al, (2005). This difference may be attributed to the discrepancies in the model structure, upper boundary calculation, and the numerical schemes used.

Table A1 demonstrated that the variations of indexes among the 5 cases are small. As mentioned above, $\alpha_k$ is not equal to one in cases 1, 2, and 4, while it is equal to one in cases 3 and 5. Thus, cases 1, 2, and 4 can be grouped together as group 1, while cases 3 and 5 can be grouped together as group 2. The differences of inner group are much smaller than inter group. This suggests that MOIST is not sensitive to the formulations used to calculate the kinetic fractionation coefficient, but rather to the consideration of kinetic fractionation itself. If kinetic fractionation is not considered ($\alpha_k$ is 1), the maximum $\delta$ values of isotope species will be underestimated, as observed in cases 3 and 5, because the isotope enrichment at soil surface will not occur. The kinetic fractionation cannot be ignored in reality. Therefore, MOIST should perform consistently across available formulations used to calculate the kinetic fractionation coefficient.

Table A1. Comparison of formulations used to calculate kinetic fractionation coefficient under non-saturated and non-isothermal conditions.

| | Case 1 | Case 2 | Case 3 | Case 4 | Case 5 |
|---|---|---|---|---|---|
| Calculated max $\delta^2H$ (‰) | 39.65 | 39.51 | 36.06 | 38.39 | 36.90 |
| Calculated max $\delta^{18}O$ (‰) | 20.84 | 20.69 | 17.12 | 19.47 | 18.05 |
| Depth of maximum (m) | 0.02 | 0.02 | 0.02 | 0.02 | 0.02 |
| Calculated liquid HDO/$H_2^{18}O$ slope | 1.93 | 1.93 | 2.13 | 2.00 | 2.07 |
| Calculated vapor HDO/$H_2^{18}O$ slope | 3.51 | 3.47 | 2.87 | 3.64 | 3.00 |

**Appendix B.**

The transport equations of soil water, heat, and isotopes are:

$$\frac{\partial \theta}{\partial t} + \frac{\partial \theta_v}{\partial t} = -\frac{\partial q}{\partial z} - S \tag{B1}$$

$$C_{soil}\frac{\partial T}{\partial t} + \rho\lambda_E\frac{\partial \theta_v}{\partial t} = -\frac{\partial q_T}{\partial z} \tag{B2}$$

$$\frac{\partial(C_{il}(\theta+\alpha\theta_v))}{\partial t} = -\frac{\partial q_i}{\partial z} - C_{il}S \tag{B3}$$

where $\theta$ and $\theta_v$ are the soil water content and equivalent liquid water content ($m^3$ $m^{-3}$), respectively; $q$ is the water flux (m s$^{-1}$); $S$ is the sink term (s$^{-1}$); $C_{soil}$ is the soil heat capacity (J m$^{-3}$ K); $T$ is the temperature; $\lambda_E$ is the latent heat of vaporization (J kg$^{-1}$); $\rho$ is the water density (kg m$^{-3}$); $q_T$ is the heat flux (J m$^{-2}$ s$^{-1}$); $C_{il}$ is the isotopic concentration of soil water (kg m$^{-3}$); $\alpha$ is the equilibrium fractionation coefficient ($\alpha^*$ in the manuscript); $q_i$ is the isotopic flux (kg m$^{-2}$ s$^{-1}$). Explanation of Equation. (B2) can be referred to Appendix C.

The equivalent liquid water content $\theta_v$ $\left(m^3_{liquid\ water}\ m^{-3}_{soil}\right)$ can be expressed by pore space within soil $\theta_s$-$\theta$ $\left(m^3_{air}\ m^{-3}_{soil}\right)$ and the saturated vapor concentration within soil air is expressed as $Cv_{sat}$ $\left(m^3_{liquid\ water}\ m^{-3}_{air}\right)$:

$$Cv_{sat} = \frac{m^3_{liquid\ water}}{m^3_{air}} = \frac{\frac{mass_{liquid\ water}}{\rho_{liquid\ water}}}{\frac{mass_{air}}{\rho_{air}}} = \frac{\frac{mass_{vapor}}{\rho_{liquid\ water}}}{\frac{mass_{air}}{\rho_{air}}} \tag{B4}$$

where $\theta_s$ is the saturated soil water content (m$^3$ m$^{-3}$); $m^3_{liquid\ water}$ and $m^3_{air}$ are the volume of liquid water (m$^3$) and air (m$^3$) within soil pore space; $mass_{liquid\ water}$ and $mass_{air}$ are the mass of liquid water (kg) and mass of air (kg) in the soil pore space; $\rho_{liquid\ water}$ and $\rho_{air}$ are the density of liquid water (kg m$^{-3}$) and air (kg m$^{-3}$), respectively; $mass_{vapor}$ is the mass of vapor (kg) within soil pore space.

Then, the ideal gas law can be incorporated into Equation. B4:

$$Cv_{sat} = \frac{\frac{mass_{vapor}}{\rho_{liquid\ water}}}{\frac{mass_{air}}{\rho_{air}}} = \frac{\rho_{air}}{\rho_{liquid\ water}}\frac{P_{vapor_{sat}}M_{water}}{P_{air}M_{air}} \tag{B5}$$

where $P_{vapor_{sat}}$ and $P_{air}$ are the saturated vapor pressure (kpa) and air pressure (kpa), respectively; $M_{water}$ and $M_{air}$ are the mole weight of water (kg mol$^{-1}$) and air (kg mol$^{-1}$), respectively.

Equation. A5 can be further simplified by applying the ideal gas law again on $P_{air}$:

$$Cv_{sat} = \frac{\rho_{air}}{\rho_{liquid\ water}}\frac{P_{vapor_{sat}}M_{water}}{P_{air}M_{air}} = \frac{\rho_{air}}{\rho_{liquid\ water}}\frac{P_{vapor_{sat}}M_{water}}{\rho_{air}\frac{R}{M_{air}}T M_{air}} \tag{B6}$$

where $R$ is the ideal gas constant (J mol$^{-1}$ K$^{-1}$). Similarly, the unsaturated vapor concentration in soil pore space in terms of equivalent liquid water content, $Cv$, is given by:

$$Cv = \frac{\rho_{air}}{\rho_{liquid\ water}}\frac{P_{vapor_{sat}}M_{water}}{P_{air}M_{air}} = \frac{\rho_{air}}{\rho_{liquid\ water}}\frac{P_{vapor}M_{water}}{\rho_{air}\frac{R}{M_{air}}T M_{air}} \tag{B7}$$

Saturated vapor pressure, $P_{vapor_{sat}}$, can be calculated by Tetens formula (Ham, 2015). Then, $Cv_{sat}$

is written as:

$$Cv_{sat} = \frac{0.61078 e^{\frac{17.269T}{T+237.29}} M_{water}}{\rho_{liquid\,water} RT}$$ (B8)

Considering the influence of variation of both soil water content and temperature on the relative humidity, $h_r$ is given by (Philip, 1957):

$$h_r = e^{\frac{M_w hg}{RT}}$$ (B9)

Besides, according to the definition of $h_r$:

$$h_r = \frac{P_{vapor}}{P_{vapor_{sat}}}$$ (B10)

which can be rewritten by combining Equations. B6 and B7:

$$h_r = \frac{Cv}{Cv_{sat}}$$ (B11)

Therefore, the equivalent water content of the volumetric water vapor content, $\theta_v$, can be written as:

$$\theta_v = (\theta_s - \theta) Cv_{sat} h_r$$ (B12)

Introducing Equation.B11 to Equations. B1 - B3:

$$\frac{\partial \theta}{\partial t} + \frac{\partial ((\theta_s - \theta)\, Cv_{sat} h_r)}{\partial t} = -\frac{\partial q}{\partial z} - S$$ (B13)

$$\frac{C_{soil} \partial T}{\partial t} + \frac{\rho \lambda_E \partial ((\theta_s - \theta) Cv_{sat} h_r)}{\partial t} = -\frac{\partial q_T}{\partial z}$$ (B14)

$$\frac{\partial \left( C_{il}(\theta + \alpha Cv_{sat} h_r (\theta_s - \theta)) \right)}{\partial t} = -\frac{\partial q_i}{\partial z} - C_{il} S$$ (B15)

Note that the head-based Richards' equation is used in our model, $\frac{\partial h}{\partial t}$, $\frac{\partial T}{\partial t}$, and $\frac{\partial C_{il}}{\partial t}$ are isolated to solve Equation. B13, B14, and B15 for $h$, $T$, and $C_{il}$ at each time step simultaneously. Since $Cv_{sat}$ is the function of $T$ (Equation. B8), $h_r$ is the function of $h$ and $T$ (Equation. B9), and $\alpha$ is the function of $T$, the analytical expressions of $\frac{\partial h}{\partial t}$, $\frac{\partial T}{\partial t}$, and $\frac{\partial C_{il}}{\partial t}$ can be written as:

$$\frac{\partial h}{\partial t} = \frac{1}{A} \left( -\frac{\partial q_T}{\partial z} - B \frac{\left( -\frac{A}{C}\left(\frac{\partial q}{\partial z} + S\right) + \frac{\partial q_T}{\partial z} \right)}{\frac{D}{C} A - B} \right)$$ (B16)

$$\frac{\partial T}{\partial t} = \frac{-\frac{A}{C}\left(\frac{\partial q}{\partial z} + S\right) + \frac{\partial q_T}{\partial z}}{\frac{D}{C} A - B}$$ (B17)

$$\frac{\partial C_{il}}{\partial t} = \frac{F}{E}$$ (B18)

with coefficients A to E:

$$A = \rho \lambda_E \left( (\theta_s - \theta)\, Cv_{sat} \frac{\partial h_r}{\partial h} - Cv_{sat} h_r \frac{\partial \theta}{\partial h} \right)$$ (B19)

$$B = C_{soil} + \rho \lambda_E \left( (\theta_s - \theta) Cv_{sat} \frac{\partial h_r}{\partial T} + (\theta_s - \theta) h_r \frac{\partial Cv_{sat}}{\partial T} \right)$$ (B20)

$$C=(1\text{-}Cv_{sat}h_r)\frac{\partial\theta}{\partial h}+(\theta_s\text{-}\theta)Cv_{sat}\frac{\partial h_r}{\partial h} \tag{B21}$$

$$D=(\theta_s\text{-}\theta)Cv_{sat}\frac{\partial h_r}{\partial T}+(\theta_s\text{-}\theta)\,h_r\frac{\partial Cv_{sat}}{\partial T} \tag{B22}$$

$$E=\theta+\alpha Cv_{sat}\,h_r\,(\theta_s\text{-}\theta) \tag{B23}$$

$$F=-\frac{\partial q_i}{\partial z}-C_{il}S-C_{il}\frac{\partial\theta}{\partial h}\frac{\partial h}{\partial t}-$$

$$C_{il}\theta_s\left(Cv_{sat}h_r\frac{\partial\alpha}{\partial T}\frac{\partial T}{\partial t}+\alpha h_r\frac{\partial Cv_{sat}}{\partial T}\frac{\partial T}{\partial t}+\alpha Cv_{sat}\left(\frac{\partial h_r}{\partial h}\frac{\partial h}{\partial t}+\frac{\partial h_r}{\partial T}\frac{\partial T}{\partial t}\right)\right)+C_{il}\left(Cv_{sat}h_r\theta\frac{\partial\alpha}{\partial T}\frac{\partial T}{\partial t}+\alpha h_r\theta\right.$$

$$\left.\frac{\partial Cv_{sat}}{\partial T}\frac{\partial T}{\partial t}+\alpha Cv_{sat}\theta\left(\frac{\partial h_r}{\partial h}\frac{\partial h}{\partial t}+\frac{\partial h_r}{\partial T}\frac{\partial T}{\partial t}\right)+\alpha Cv_{sat}h_r\frac{\partial\theta}{\partial h}\frac{\partial h}{\partial t}\right) \tag{B24}$$

Equations. B13, B14, and B15 were transformed into a system of coupled ordinary differential equations by Equations. B16-B24. This system is solved by MATLAB solvers (ode113/ode23tb) simultaneously. The derivative vector ($\frac{\partial h}{\partial t}$, $\frac{\partial T}{\partial t}$, and $\frac{\partial C_{il}}{\partial t}$), having a length of the number of spatial discretization multiplied by three.

To construct the derivative vector, values from Equation. B17 were calculated firstly because they were also used in Equation. B16. Equation. B17 showed that temporal variation of temperature was influenced by $q$, $q_T$, $\theta$, and other parameters from coefficients *A-D*. Reversely, *T* influences $h_r$ and $Cv_{sat}$ (Equations. B19-B22) and further affect the water transport within soil (Equation. B16).

Equation. B16 showed that temporal variation of *h* was closely related to *T* because $\frac{\left(-\frac{A}{C}\left(\frac{\partial q}{\partial z}+S\right)+\frac{\partial q_T}{\partial z}\right)}{\frac{D}{C}A\text{-}B}$ from Equation. B16 is $\frac{\partial T}{\partial t}$ (Equation. B17). Besides, soil heat properties, such as soil heat capacity and latent heat of vaporization (included in coefficients *A* and *B*), were also involved in soil water (vapor) movement.

Equation. B24 shows that isotope transport was influenced by all the parameters coupled. Specifically, water transport (Equation. B16) affects isotopic fluxes since isotopes were treated as solutes, while heat transport (Equation. B17) had an influence on equilibrium fractionation coefficients and further on the isotopic concentration in soil water. Both water and heat transport affected $h_r$ and $Cv_{sat}$ in soil. Therefore, values for derivative vector construction from Equation. B17 were calculated based on Equations. B16, B17, B23, and B24.

Finally, the derivative vector, along with the initial conditions and the time span were passed to the solvers. The solver then computed the solution of this system over the specified time span numerically. Numerical schemes of solvers are described below.

*Numerical scheme of ode113*
The ode113 solver uses an adaptive, variable-order, variable-step-size (VOVS) method. This is implemented with a variable order Adams-Bashforth-Moulton (ABM) method (ode113, 2023),

which is a combination of an explicit types of the Adams-Bashforth (AB) and an implicit type of Adams-Moulton (AM) methods. Specifically, the AB method is used to estimate the solution at the new time step by taking multiple previous time steps into account, while the AM method is used to make corrections.

The ode113 can select automatically between the $1^{st}$ and $13^{th}$ order approximation (the highest order used appears to be 12) during the computation based on the estimation errors. This is helpful for minimizing the estimated errors and for achieving high efficiency in time. Moreover, the time step size is adjusted according to the estimation error. In this way, ode113 can handle a wide range of ODE problems with high accuracy and efficiency.

Therefore, ode113 can do a good job when the transport media is relatively uniform. However, ode113 is susceptible to numerical oscillation when hydraulic conductivities between layers differed greatly because of the adopted explicit scheme.

*Numerical schemes of ode23tb*
Ode23tb is a solver specifically designed for solving ODEs with highly oscillatory solutions (ode23tb, 2023), such as those arising from heterogeneity in hydraulic conductivities between soil layers. The 'tb' stands for that the solver combines a trapezoidal rule (sometimes referred as the second-order AM method (Adams methods, 2023)) with a $2^{nd}$ order backward differentiation formula (BDF). Because of this, ode23tb is an efficient and accurate solver for stiff ODE systems, making it less susceptible to numerical instability.

Like ode113, ode23tb can adjust the step size automatically based on the estimated error and the oscillatory behavior of the solution. However, ode23tb is an implicit solver, making it more computationally expensive than other solvers. But because it adopts the trapezoidal BDF method, it is more efficient and accurate than other types of implicit methods, such as the fully implicit Euler method or the backward Euler method (Time integration, 2023). Therefore, ode23tb may work better than ode113 when the soil physical properties are greatly differed between layers.

**Appendix C.**

The heat transport equation within soil is written as:

$$C_{soil}\frac{\partial T}{\partial t} + \rho\lambda_E\frac{\partial \theta_v}{\partial t} = \frac{\partial}{\partial z}\left(k_H\frac{\partial T}{\partial z}\right) - C_w q_l\frac{\partial T}{\partial z} - C_{vh}\frac{\partial q_v T}{\partial z} - \rho\lambda_E\frac{\partial q_v}{\partial z} \tag{C1}$$

Equation. (C1) can be rewritten by the chains rule:

$$C_{soil}\frac{\partial T}{\partial t} + \rho\lambda_E\frac{\partial \theta_v}{\partial t} = \frac{\partial}{\partial z}\left(k_H\frac{\partial T}{\partial z}\right) - C_w q_l\frac{\partial T}{\partial z} - C_{vh}q_v\frac{\partial T}{\partial z} - C_{vh}T\frac{\partial q_v}{\partial z} - \rho\lambda_E\frac{\partial q_v}{\partial z} \tag{C2}$$

Then:

$$C_{soil}\frac{\partial T}{\partial t} + \rho\lambda_E\frac{\partial \theta_v}{\partial t} = \frac{\partial}{\partial z}\left(k_H\frac{\partial T}{\partial z}\right) - (C_w q_l + C_{vh}q_v)\frac{\partial T}{\partial z} - (C_{vh}T + \rho\lambda_E)\frac{\partial q_v}{\partial z} \tag{C3}$$

where $C_w q_l + C_{vh}q_v$ and $C_{vh}T + \rho\lambda_E$ can be treated as constants within each layer and Equation. (C3) is rewritten as:

$$C_{soil}\frac{\partial T}{\partial t} + \rho\lambda_E\frac{\partial \theta_v}{\partial t} = -\frac{\partial}{\partial z}\left(-k_H\frac{\partial T}{\partial z} + Constant_1 T + Constant_2 q_v\right) \tag{C4}$$

where $Constant_1 = C_w q_l + C_{vh}q_v$ and $Constant_2 = C_{vh}T + \rho\lambda_E$.

Assuming $q_T = -k_H\frac{\partial T}{\partial z} + Constant_1 T + Constant_2 q_v$, Equation (C4) can be further simplified:

$$C_{soil}\frac{\partial T}{\partial t} + \rho\lambda_E\frac{\partial \theta_v}{\partial t} = -\frac{\partial q_T}{\partial z} \tag{C5}$$

Equation. (C5) is the same as Equation. (B2).

**Reference**

Braud, I., Bariac, T., Gaudet, J. P., and Vauclin, M.: SiSPAT-Isotope, a coupled heat, water and stable isotope (HDO and H 218O) transport model for bare soil. Part I. Model description and first verifications, J Hydrol (Amst), 309, 277–300, https://doi.org/10.1016/j.jhydrol.2004.12.013, 2005.

Brutsaert, W.: Evaporation into the atmosphere, theory, history and applications, Springer Dordrecht, Netherlands, 302 pp., https://doi.org/10.1007/978-94-017-1497-6, 1982.

Time integration:
https://space.mit.edu/RADIO/CST_online/mergedProjects/DES/theory/time_integration.htm, last access: 28 February 2023.

Ham, J. M.: Useful equations and tables in micrometeorology, Micrometeorology in Agricultural Systems, 533–560, https://doi.org/10.2134/agronmonogr47.c23, 2015.

Haverd, V. and Cuntz, M.: Soil-Litter-Iso: A one-dimensional model for coupled transport of heat, water and stable isotopes in soil with a litter layer and root extraction, J Hydrol (Amst), 388, 438–455, https://doi.org/10.1016/j.jhydrol.2010.05.029, 2010.

ode23tb: https://www.mathworks.com/help/matlab/ref/ode23tb.html, last access: 28 February 2023.

ode113: https://www.mathworks.com/help/matlab/ref/ode113.html, last access: 28 February 2023.

Melayah, A., Bruckler, L., and Bariac, T.: Modeling the transport of water stable isotopes in unsaturated soils under natural conditions 1. Theory, Water Resour Res, 32, 2047–2054, https://doi.org/10.1029/96WR00674, 1996.

Philip, J. R.: Evaporation, and moisture and heat fields in the soil, Journal of Meteorology, 14, 354–366, https://doi.org/https://doi.org/10.1175/1520-0469(1957)014<0354:EAMAHF>2.0.CO;2, 1959.

Stumpp, C., Stichler, W., Kandolf, M., and Šimůnek, J.: Effects of Land Cover and Fertilization Method on Water Flow and Solute Transport in Five Lysimeters: A Long-Term Study Using Stable Water Isotopes, Vadose Zone Journal, 11, https://doi.org/10.2136/vzj2011.0075, 2012.

Adams methods:
https://web.mit.edu/10.001/Web/Course_Notes/Differential_Equations_Notes/node6.html, last access: 28 February 2023.

Zhou, T., Šimůnek, J., and Braud, I.: Adapting HYDRUS-1D to simulate the transport of soil water isotopes with evaporation fractionation, Environmental Modelling and Software, 143, https://doi.org/10.1016/j.envsoft.2021.105118, 2021.